# Autophagy-enhancing *ATG16L1* polymorphism is associated with improved clinical outcome and T-cell immunity in chronic HIV-1 infection

Renée R. C. E. Schreurs[1,2], Athanasios Koulis[1,2], Thijs Booiman [1,2], Brigitte Boeser-Nunnink[1,2], Alexandra P. M. Cloherty[1,2], Anusca G. Rader [1,2], Kharishma S. Patel[1,2], Neeltje A. Kootstra [1,2] & Carla M. S. Ribeiro [1,2] ✉

Chronic HIV-1 infection is characterized by T-cell dysregulation that is partly restored by antiretroviral therapy. Autophagy is a critical regulator of T-cell function. Here, we demonstrate a protective role for autophagy in HIV-1 disease pathogenesis. Targeted analysis of genetic variation in core autophagy gene *ATG16L1* reveals the previously unidentified rs6861 polymorphism, which correlates functionally with enhanced autophagy and clinically with improved survival of untreated HIV-1-infected individuals. T-cells carrying *ATG16L1* rs6861(TT) genotype display improved antiviral immunity, evidenced by increased proliferation, revamped immune responsiveness, and suppressed exhaustion/immunosenescence features. In-depth flow-cytometric and transcriptional profiling reveal T-helper-cell-signatures unique to rs6861(TT) individuals with enriched regulation of pro-inflammatory networks and skewing towards immunoregulatory phenotype. Therapeutic enhancement of autophagy recapitulates the rs6861(TT)-associated T-cell traits in non-carriers. These data underscore the in vivo relevance of autophagy for longer-lasting T-cell-mediated HIV-1 control, with implications towards development of host-directed antivirals targeting autophagy to restore immune function in chronic HIV-1 infection.

Human immunodeficiency virus-1 (HIV-1) establishes chronic infection, which is characterized by persistent viral replication, systemic decline in T helper (Th) cell numbers, and accumulation of immunologic defects[1]. The introduction of combination antiretroviral therapy (cART) ushered in a new era for HIV-1 infected individuals with high levels of efficacy in suppressing viral replication and transmission as well as safety and tolerability in people living with HIV[1,2]. However, while cART effectively reduces viral burden and prolongs life expectancy, it is not curative. cART-treated HIV-1-infected individuals exhibit persistent immune activation and inflammation leading to an increased risk of developing non-AIDS comorbidities such as cardiovascular and neurocognitive diseases[3,4]. In order to circumvent the limitations of cART and promote immune control, alternative combinatory HIV-1 therapeutic strategies are imperative.

One promising approach to develop new anti-HIV-1 therapeutics is to capitalize on natural host defense against HIV-1 such as host restriction factors and antiviral pathways that interfere with viral replication and transmission. Notably, we were the first to demonstrate

[1]Amsterdam UMC location University of Amsterdam, Experimental Immunology, Meibergdreef 9, Amsterdam, The Netherlands. [2]Amsterdam institute for Immunology & Infectious Diseases, Amsterdam, The Netherlands. ✉e-mail: c.m.ribeiro@amsterdamumc.nl

that the host restriction factor tripartite motif protein 5 alpha isoform (TRIM5α) restricts HIV-1 infection via an autophagy-mediated mechanism in a specific subset of primary human dendritic cells (DCs)[5]. Others have described the antiviral function of human TRIM5α in HIV-1 infection of CD4+ T lymphocytes and during the clinical course of HIV-1 infection[6–9]. Furthermore, HIV-1-infected individuals with spontaneous control of HIV-1 (long-term nonprogressors and elite controllers) display increased numbers of autophagy vesicles and higher TRIM5α expression levels in peripheral blood mononuclear cells (PBMCs) as compared to progressors[8,10,11], suggesting an important role for host autophagy mechanisms in controlling the disease progression of treatment-naïve HIV-1 infected individuals.

Autophagy refers to the process of macro-autophagy, a 'self-degradative' process whereby cytosolic constituents are captured in double membrane-bound vesicles, called autophagosomes, destined for degradation[12]. Autophagy is a multi-step process regulated by mechanistic target of rapamycin (mTOR; inhibitor) and activatorAMP-activated kinase (AMPK; activator). Initiation of autophagy is triggered by reversal of mTOR's inhibition of autophagy after which autophagosomes are synthesized and elongated by the coordinated activity of autophagy-related (ATG) proteins[13]. Mature autophagosomes then fuse with lysosomes resulting in degradation of the cargo. One such ATG protein is ATG16L1, which is a key mediator in autophagy processes with multifaceted activities from autophagosome biogenesis and cargo-recruitment to control of inflammatory processes[14,15]. Notably, we have previously shown that autophagy suppresses mucosal HIV-1 infection and transmission at the early steps of the virus replication cycle via an ATG16L1-TRIM5α-mediated mechanism[5,16,17].

Autophagy has emerged as a crucial regulator of T-cell functionality including proliferation, activation and differentiation[18]. Genetic variation in core autophagy gene *ATG16L1* has been implicated in inflammatory immune disorders; a single nucleotide polymorphism (SNP) in *ATG16L1* resulting in a threonine-to-alanine substitution at position 300 (T300A; rs2241880) is associated with decreased autophagy and increased susceptibility to Crohn's disease[19,20]. The autophagy-deficient *ATG16L1* T300A variant also impacts CD4+ T helper cell responses in individuals suffering from Crohn's disease and has been associated with concomitant loss of regulatory T-cell (Treg) responses and increased frequencies of inflammatory Th1 and Th17 cells in murine intestines[21,22], underscoring the significance of functional autophagy in curbing Th17-mediated inflammatory disorders.

The T-cell immune response is a tightly regulated process consisting of antigen recognition by the TCR, co-stimulation (via CD27 and CD28), and a self-limiting termination signal provided by inhibitory immune checkpoints (ICs) such as programmed cell-death 1 (PD-1) and cytotoxic T-lymphocyte-associated protein 4 (CTLA-4)[23]. Additionally, the magnitude of the T-cell response is mediated via interleukin (IL)−2-(CD25) and IL-7-receptor (CD127) signaling[24]. Under normal circumstances, the expression of ICs declines to normal levels after pathogen clearance. However, during chronic HIV-1 infection, T cells eventually enter a dysfunctional state called exhaustion, a progressive condition that leads to sub-optimal immune responsiveness and increased expression of ICs, allowing manifestation of opportunistic infections that rarely occur in immune-competent individuals[25,26]. Notably, exhausted T cells are characterized by defective autophagy[27,28]. cART treatment induces durable suppression of plasma viremia and supports gradual CD4+ T cell recovery in HIV-1-infected individuals. Nevertheless, effectively treated HIV-1-infected individuals, who display high levels of immune activation in the absence of overt infection ('sterile inflammation') or failing to achieve complete immune reconstitution, are at higher risk to develop immune deficiency-associated pathologies[29,30].

Autophagy activity not only controls acute HIV-1 infection, but also plays a key role in HIV-1 disease pathogenesis. Increased numbers of autophagy vesicles have been implicated in resistance to HIV-1 disease progression[11], however, the specific contribution of autophagy molecules to immunological and clinical responses in vivo remains unclear. Here, we demonstrate that the rs6861 genetic polymorphism in critical autophagy gene *ATG16L1* impacts the natural course of untreated HIV-1 chronic disease. Notably, our study shows that HIV-1-infected individuals carrying the *ATG16L1* rs6861(TT) genotype−i.e. homozygous for the minor allele−display prolonged survival and delayed disease progression. Notably, in-depth phenotypic and functional flow-cytometric analyses in rs6861(TT) genotyped HIV-1 infected individuals revealed augmented T-cell effector responses accompanied by higher proliferation potential, superior immune responsiveness and decreased exhaustion signatures. Transcriptomic analyses in combination with functional Th subset profiling indicated increased capacity for immune regulation and reduced Th17 inflammatory responses in rs6861(TT) genotyped individuals. Furthermore, treatment of PBMCs derived from *ATG16L1* rs6861(CC) genotype donors−i.e. homozygous for the major allele−with autophagy-enhancing therapeutics reprogrammed T cells to exhibit the phenotypic signatures observed in rs6861(TT) genotyped T-cells. Our study provides new evidence on the in vivo relevance of autophagy in containment of HIV-1 disease pathogenesis, thus delineating autophagy-modulating interventions as promising host-directed strategies to further advance cART therapies towards boosting T-cell functionality and mitigating immune activation cascades in chronically HIV-1-infected individuals.

## Results

### Genetic variation in core autophagy gene *ATG16L1* impacts HIV-1 disease pathogenesis

We have previously shown that the autophagy molecule ATG16L1 forms an autophagy-activating scaffold with human TRIM5α, which targets HIV-1 capsid complexes for degradation resulting in autophagy-mediated restriction of mucosal HIV-1 infection[5]. Additionally, abundance of autophagy vesicles has been implicated in the containment of disease pathogenesis in HIV-1 controllers and different human *TRIM5* polymorphisms have been shown to impact the clinical course of HIV-1 infection[8,10,11]. Here, we set out to investigate the role of autophagy in the natural course of HIV-1 infection. To this end, we executed a targeted analysis of tagging SNPs in core autophagy gene *ATG16L1* using available data from 304 untreated HIV-1-infected men-who-have-sex-with-men (MSM), including genotyping analyses performed with Illumina's Infinium HumanHap 300 BeadChip, within the Amsterdam Cohort Studies (ACS) on HIV infection and AIDS (Fig. 1a, b)[31]. For four out of the nine *ATG16L1* tagging SNPs within the ACS cohort, less than 10 participants homozygous for the less common allele of these SNPs were available, and therefore excluded from subsequent analyses (Fig. 1c−gray shaded). Univariate Cox regression survival analysis for the remaining five *ATG16L1* SNPs revealed a significant protective effect for two *ATG16L1* SNPs (rs6861 and rs7563345) using AIDS diagnosis (Center for Disease Control [CDC] 1987) as endpoint, and for four *ATG16L1* SNPs (rs2241880, rs6861, rs3792106 and rs7563345) using AIDS-related death as endpoint (Fig. 1d). The protective effect of *ATG16L1* SNP rs6861, but not the others, on both endpoints remained significant after Bonferroni multiple testing correction (adjusted $p < 0.01$). *ATG16L1* SNP rs6861 was thereby identified to be strongly associated with delayed HIV-1 disease progression (Fig. 1d, e). Of note, the protective effect observed for *ATG16L1* rs6861 polymorphism on HIV-1 disease pathogenesis was independent of both CCR5Δ32 and HLA-B57 genotypes (Supplementary Table 1)[32–35].

### *ATG16L1* rs6861(TT) genotype is associated with prolonged control of HIV-1 disease pathogenesis

Kaplan−Meier survival analyses of *ATG16L1* rs6861 genetic variants demonstrated that HIV-1-infected individuals carrying the rs6861(TT) genotype−i.e. homozygous for the minor/alternate allele−displayed

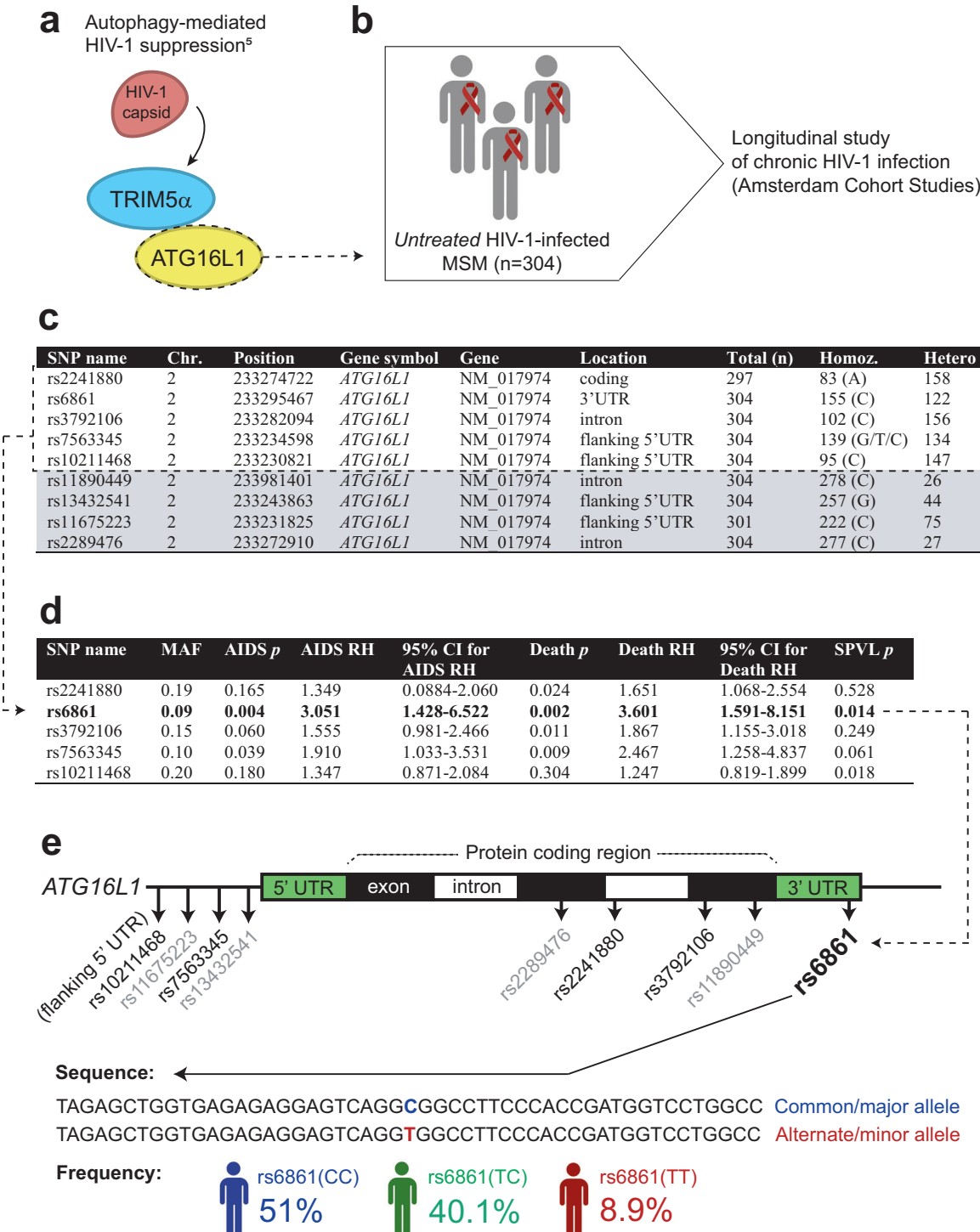

**c**

| SNP name | Chr. | Position | Gene symbol | Gene | Location | Total (n) | Homoz. | Hetero | Homoz. |
|---|---|---|---|---|---|---|---|---|---|
| rs2241880 | 2 | 233274722 | *ATG16L1* | NM_017974 | coding | 297 | 83 (A) | 158 | 56 (G) |
| rs6861 | 2 | 233295467 | *ATG16L1* | NM_017974 | 3'UTR | 304 | 155 (C) | 122 | 27 (T) |
| rs3792106 | 2 | 233282094 | *ATG16L1* | NM_017974 | intron | 304 | 102 (C) | 156 | 46 (T) |
| rs7563345 | 2 | 233234598 | *ATG16L1* | NM_017974 | flanking 5'UTR | 304 | 139 (G/T/C) | 134 | 31 (A) |
| rs10211468 | 2 | 233230821 | *ATG16L1* | NM_017974 | flanking 5'UTR | 304 | 95 (C) | 147 | 62 (T/G) |
| rs11890449 | 2 | 233981401 | *ATG16L1* | NM_017974 | intron | 304 | 278 (C) | 26 | 0 (T) |
| rs13432541 | 2 | 233243863 | *ATG16L1* | NM_017974 | flanking 5'UTR | 304 | 257 (G) | 44 | 3 (A) |
| rs11675223 | 2 | 233231825 | *ATG16L1* | NM_017974 | flanking 5'UTR | 301 | 222 (C) | 75 | 4 (A) |
| rs2289476 | 2 | 233272910 | *ATG16L1* | NM_017974 | intron | 304 | 277 (C) | 27 | 0 (A/G/T) |

**d**

| SNP name | MAF | AIDS *p* | AIDS RH | 95% CI for AIDS RH | Death *p* | Death RH | 95% CI for Death RH | SPVL *p* |
|---|---|---|---|---|---|---|---|---|
| rs2241880 | 0.19 | 0.165 | 1.349 | 0.0884-2.060 | 0.024 | 1.651 | 1.068-2.554 | 0.528 |
| **rs6861** | **0.09** | **0.004** | **3.051** | **1.428-6.522** | **0.002** | **3.601** | **1.591-8.151** | **0.014** |
| rs3792106 | 0.15 | 0.060 | 1.555 | 0.981-2.466 | 0.011 | 1.867 | 1.155-3.018 | 0.249 |
| rs7563345 | 0.10 | 0.039 | 1.910 | 1.033-3.531 | 0.009 | 2.467 | 1.258-4.837 | 0.061 |
| rs10211468 | 0.20 | 0.180 | 1.347 | 0.871-2.084 | 0.304 | 1.247 | 0.819-1.899 | 0.018 |

**Fig. 1 | Genetic variation in key autophagy gene *ATG16L1* impacts the natural course of HIV-1 disease pathogenesis. a, b** Schematic representations of the TRIM5α−ATG16L1−p24 capsid complex involved in autophagy-mediated HIV-1 suppression (**a**)[5], and the targeted analysis of tagging SNPs in the *ATG16L1* gene within a cohort of 304 untreated HIV-1-infected MSM enrolled in the Amsterdam Cohort Studies (ACS) on HIV infection and AIDS (**b**)[31]. **c** Table depicting the nine *ATG16L1* tagging SNPs investigated within the ACS cohort; four were excluded from subsequent analyses (gray shaded) since <10 individuals homozygous for one of the alleles were available for investigation. **d** Table showing (unadjusted) *p*-values of univariate survival analyses using Cox regression of the remaining five SNPs with endpoints AIDS diagnosis (CDC 1987), AIDS-related Death, or set-point viral load (SPVL). After Bonferroni correction for multiple testing (adjusted *p* < 0.01), rs6861 (in bold) was identified to remain strongly associated with delayed HIV-1 disease progression (*p* = 0.00996). **e** Schematic representation of the nine tagging SNPs investigated within the *ATG16L1* gene, including sequence and allele frequency for rs6861. MSM men-who-have-sex-with-men, Chr. chromosome, MAF minor (less common) allele frequency, RH relative hazard, CI confidence interval, Death AIDS-related death, SPVL set-point viral load. See Supplementary Table 1. **b, e** Created with Biorender.com.

delayed disease progression, compared to rs6861(CC) and heterozygous rs6861(CT) genotypes, using AIDS diagnosis CDC 1987 (Log-rank *p*-value = 0.0024; Fig. 2a) and CDC 1993 as endpoints (Log-rank *p*-value = 0.0242; Supplementary Fig. 1). Markedly, the *ATG16L1* rs6861(TT) genotype was significantly associated with prolonged survival of untreated HIV-1-infected individuals using AIDS-related death as endpoint (Log-rank *p*-value = 0.0009; Fig. 2b). Furthermore, rs6861(TT) genotyped individuals displayed a decreased set-point plasma viral load and increased CD4 cell counts as compared to rs6861(CC) and rs6861(CT) genotyped individuals (Fig. 2c, d). Our data thereby indicate that the rs6861(TT) genetic variant in key autophagy gene *ATG16L1* is associated with delayed HIV-1 disease progression in these treatment-naïve HIV-1 infected individuals.

### rs6861(TT) genotyped CD4+ T helper cells display enhanced autophagy flux and upregulated transcription of core autophagy and lysosome genes

As autophagy, and in particular ATG16L1, play key roles in the regulation of T-cell responses[18,21], we next determined autophagy activity in combination with functional and phenotypical characteristics of the T-cell compartment in rs6861(TT) versus rs6861(CC) individuals.

To investigate whether the rs6861(TT) genetic variant affected autophagy-mediated lysosomal degradative activity (i.e. autophagy flux), PBMCs were isolated from healthy genotyped donors and intracellular levels of the canonical autophagosome-associated LC3-II protein was determined in combination with immune cell lineage markers by flow-cytometry analysis[36,37]. Both rs6861(TT) and rs6861(CC) genotypes are equipped with functional autophagy flux processes as evidenced by accumulation of intracellular LC3-II protein upon treatment with V-ATPase inhibitor bafilomycin A1 (Fig. 3a)[37].

Notably, the total lymphocyte population, and in particular CD4+ T cells derived from rs6861(TT) individuals, displayed an intrinsically higher number of autophagosomes as represented by a relative greater increment of LC3-II median fluorescence intensity (MFI) when compared to rs6861(CC) individuals. (Fig. 3a, b). In line with previous reports[38,39], TCR-mediated activation using soluble anti-CD3 and anti-CD28 potently induced autophagy flux (Fig. 3c). Furthermore, the levels of autophagy flux in response to TCR-engagement in combination with bafilomycin A1-treatment was also higher in CD4+ T cells from rs6861(TT) compared to rs6861(CC) individuals (Fig. 3c), underscoring the intrinsically higher autophagy flux potential of rs6861(TT) genotyped individuals.

TRIM5α, which interacts with different autophagy molecules including ATG16L1, has been identified as a HIV-1 restriction factor in CD4+ T cells[5,7,40]. Higher intracellular TRIM5α expression levels have also been associated with natural HIV-1 control[11,41]. A relatively small fraction of blood-derived CD4+ T cells ( ~ 3%) expressed TRIM5α (Fig. 3d). rs6861(TT) genotyped TRIM5α+ CD4+ T cells displayed an increased MFI as compared to rs6861(CC), but no difference in frequency of TRIM5α+ CD4+ T cells between the genotypes was detected (Fig. 3d, Supplementary Fig. 2a, b), suggesting increased intracellular expression of TRIM5α in CD4+ T cells carrying the rs6861(TT) genetic variant. TCR-engagement consistently resulted in an increased frequency of TRIM5α+CD4+ T cells in rs6861(TT), whereas the effect of TCR stimulation on rs6861(CC) genotyped CD4+ T cells was less robust (Supplementary Fig. 2a).

In order to gain deeper insight into the baseline autophagy status and autophagy-related gene expression, we performed next-generation sequencing (NGS) of RNA isolated from fluorescence-activated cell-sorted (FACSorted) genotyped CD4+ T cells at

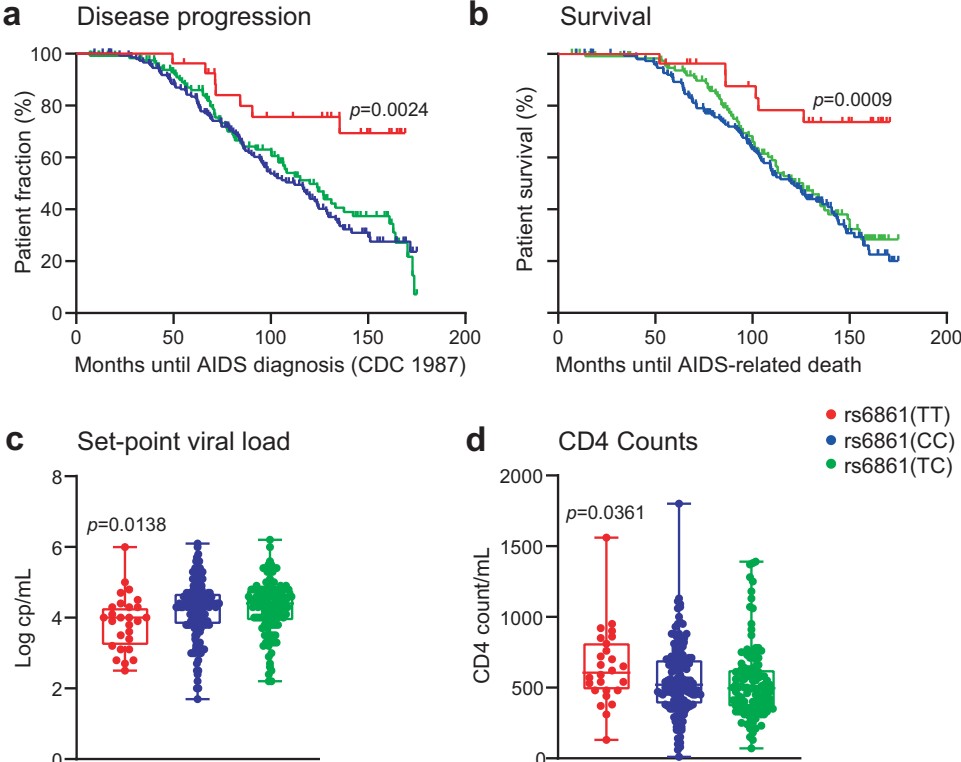

**Fig. 2 | *ATG16L1* rs6861(TT) genotype is associated with delayed disease progression, improved AIDS-free survival, decreased set-point viral load and increased CD4 counts in untreated HIV-1 infected individuals. a,b** Kaplan–Meier Survival analyses (Log-rank tests) using the endpoints AIDS diagnosis (CDC1987; **a**) and AIDS-related death (**b**). **c, d** Independent two-tailed *t*-tests comparing set-point viral load in plasma (**c**) and CD4 counts (**d**) of rs6861(TT) HIV-1-infected individuals (*n* = 27) to the combined rs6861(CC) (*n* = 122) and rs6861(CT) (*n* = 155) variants (after Eigenstrat). Minimum-to-maximum whiskers are shown with boxes indicating 25th and 75th percentile ± median. See Supplementary Fig. 1. Source data are provided as a Source Data file.

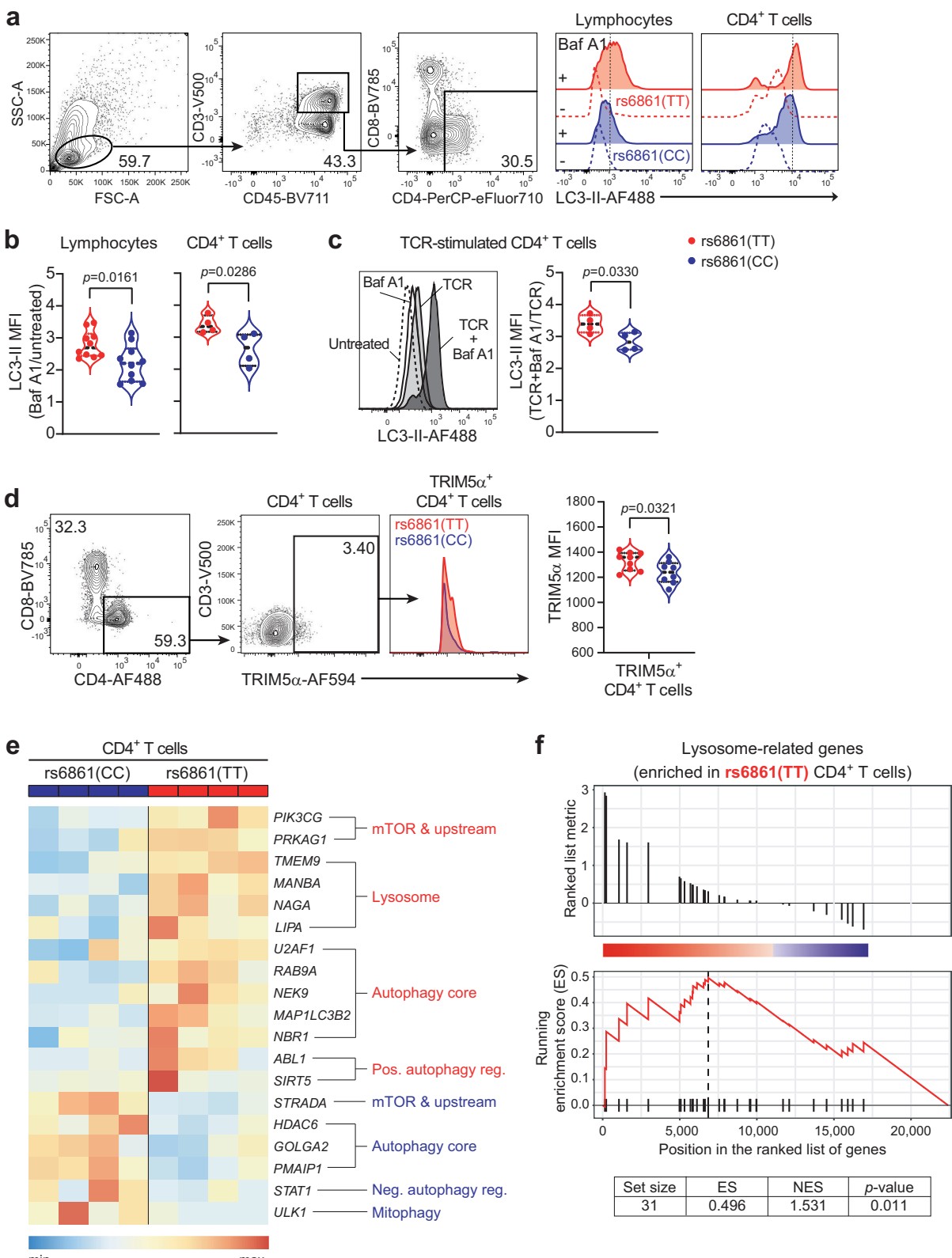

steady-state (i.e. untreated). Subsequently, a targeted analysis of the 604 annotated genes associated with the different steps of the autophagy pathway as recently compiled by Bordi et al.[42], revealed an upregulation of autophagy core genes associated with enhanced autophagy-mediated lysosomal degradation (i.e. genes related to lysosomes and positive regulation of autophagy processes) in rs6861(TT) genotyped CD4+ T cells as compared to rs6861(CC), the

latter which was used as a relative baseline for this study. Additionally, autophagy genes particularly upregulated in rs6861(CC) CD4+ T cells were mostly associated with negative regulation of autophagy processes and mitophagy (i.e removal of damaged mitochondria through autophagy) (Fig. 3e). In line with the functional LC3-II analysis (Fig. 3a, b), gene set enrichment analysis (GSEA) of the main autophagy categories assigned by Bordi et al.[42] further underscored enrichment

**Fig. 3 | rs6861(TT) T helper cells display enhanced autophagy flux and upregulated expression of autophagy molecules. a–c** Flow-cytometric representation of autophagy flux as measured by saponin extraction of LC3-II accumulation (MFI) in combination with immune cell surface markers upon incubation with bafilomycin A1 (**a**), showing (relative) increased autophagy flux in lymphocytes ($n = 10$ per genotype) and CD4+ T cells ($n = 4$ per genotype) at steady state (**b**) or upon TCR-mediated stimulation (**c**) in rs6861(TT) compared to rs6861(CC) CD4+ T cells ($n = 4$ per genotype). **d** Increased expression of TRIM5α (MFI) within rs6861(TT) TRIM5α+ CD4+ T cells (TT $n = 9$, CC $n = 8$) at steady-state (**b–d** independent two-tailed $t$-tests). **e** Targeted RNA-seq. analysis of autophagy-associated genes[42] in steady-state CD4+ T cells, indicating upregulated transcription of core autophagy genes and genes related to positive regulation of autophagy and lysosomes in rs6861(TT) compared to rs6861(CC) CD4+ T cells ($n = 4$ per genotype). **f** GSEA (Kolmogorov–Smirnov test, unadjusted $p$-values) in rs6861(TT) CD4+ T cells corroborating enrichment in lysosome-related genes ($n = 4$ per genotype). Baf A1 bafilomycin A1, MFI median fluorescence intensity, TCR anti-CD3/CD28 stimulation, pos. positive, reg. regulator, neg. negative, ES enrichment score, NES normalized enrichment score. See Supplementary Fig. 2. Source data are provided as a Source Data file. RNA-seq. data are accessible through GEO Series accession number GSE253769.

of lysosome-related genes specifically in rs6861(TT) CD4+ T cells (Fig. 3f). Thus, our findings strongly suggest that the *ATG16L1* rs6861(TT) genetic variant is associated with augmented autophagy-related lysosomal transcriptional networks accompanied by intrinsically enhanced levels of autophagy activity in CD4+ T cells.

### *ATG16L1* rs6861(TT) genotyped CD4 + T cells exhibit high proliferation potential and expanded T-helper subset compartments

Autophagy is a crucial regulator of T-cell homeostasis, activation and differentiation, and defective autophagy results in T-helper cell dysfunction[18]. Next, we investigated the phenotypical and functional characteristics of the T-helper cell compartment within genotyped individuals using high-parameter flow cytometry. Increased frequency of CD4+ T cells was detected in rs6861(TT) PBMCs derived from healthy donors (Fig. 4a, b). Furthermore, based on the expression of well-defined differentiation markers, the CD4+ T cell compartment of rs6861(TT) genotyped individuals had increased frequencies of CD45RA+CCR7+ naïve (Tn), CCR7⁻CD45RA-re-expressing effector-memory (Temra) and CCR7⁻CD45RA⁻CD25+ CD127- effector-memory (Tem) cells, indicative of enriched naïve, antiviral, and T regulatory cell compartments, respectively (Fig. 4a, b). Additionally, the frequency of TRIM5α-expressing cells was most prominent in the CD4+ Tn compartment of rs6861(TT) compared to rs6861 (CC) genotyped cells (Fig. 4c).

Next, proliferative potential was determined by division of Cell-Trace reagent among daughter cells upon TCR-engagement using soluble anti-CD3 and anti-CD28. rs6861(TT) genotyped CD4+ T cells displayed increased proliferative capacity compared to rs6861(CC) cells (Fig. 4d), which was particularly evident in the CCR7+ Tn compartment (Fig. 4e). T cells from HIV-1-infected individuals typically proliferate poorly in response to antigenic stimulation, precluding inefficient immune responsiveness to HIV-1 and other pathogens[43–45]. Notably, rs6861(TT) CD4+ T cells derived from chronically HIV-1-infected individuals (clinical characteristics displayed in Supplementary Fig. 3) also demonstrated higher proliferative capacity in response to TCR-mediated activation as well as upon stimulation with a HIV-1 peptide pool (Fig. 4f). Furthermore, a more prominent enrichment of the CCR7+ compartment was observed in TCR-activated CD4+ T cells derived from rs6861(TT) genotyped HIV-1-infected individuals (Fig. 4g).

Next, to further substantiate the beneficial effect of enhanced autophagy activity on T-cell functionality, we investigated the potential of targeting autophagy to improve T cell proliferation in rs6861(CC) genotyped individuals. To this end, three autophagy-enhancing drugs, which are either approved by the U.S. Food and Drug Administration (FDA) or in clinical trial, were selected based on our and others' previous research: carbamazepine, everolimus, and MK-2206[46–48]. Each drug increases autophagy flux by targeting either mTOR Complex 1/2 (mTORC1/2) or Akt molecules (Supplementary Fig. 4a). Enhanced autophagy upon treatment with the three drugs was confirmed in a U87.LC3-mCherry-GFP reporter cells (Supplementary Fig. 4b). Viability staining of genotyped PBMC treated with a range of concentrations of the drugs with autophagy-enhancing potential

indicated the drugs were not toxic at any concentration tested (Supplementary Fig. 4c). Additionally, evaluation of T-cell proliferation in response to treatment with the three autophagy-enhancing therapeutics indicated concentration-independent effects (within the range tested) on genotyped donors (Supplementary Fig. 4d). Median concentrations within the tested range were therefore selected: 100 uM carbamazepine, 5 nM everolimus, and 1 uM MK-2206. Notably, all three tested therapeutics further increased proliferation of rs6861(CC) genotyped CD4+ T cells as compared to TCR-only activated T cells (Fig. 4h). Treatment with everolimus and MK-2206 did not further increase proliferation of rs6861(TT) CD4+ T cells (Supplementary Figs. 4d and 5a). Moreover, particularly the CCR7+ compartment was enriched in response to MK-2206 in rs6861(CC) but not rs6861(TT) CD4+ T cells (Fig. 4i, Supplementary Fig. 5b), which further underpins the inherent higher levels of basal autophagy activity in these genotyped individuals (Supplementary Fig. 5a, b). These data demonstrate that intrinsically or pharmaceutically enhanced levels of autophagy flux is associated with augmented CD4+ T-cell responses with enhanced proliferative capacity and renewal of the lymph-node homing CD4+ T-cell compartment.

### CD4+ T cells derived from rs6861(TT) genotyped HIV-1-infected individuals display enhanced immune responsiveness and reduced exhaustion signatures

Sustained immune activation and increased T-cell exhaustion are hallmarks of chronic HIV-1 infection[3,25,49]. The T-cell compartment in HIV-1-infected individuals is typically characterized by a low CD4:CD8 ratio, depletion of CCR7+ naïve cells, and increased turnover (Ki-67-expression) in combination with poor proliferative capacity[50–52]. Our results demonstrated that rs6861(TT) genotyped HIV-1-infected individuals during chronic infection displayed increased CD4 counts (Fig. 2d) and CCR7+ CD4+ T cells, as well as enhanced proliferative capacity (Fig. 4f, g) as compared to rs6861(CC). CD4+ T cells from HIV-1-infected individuals typically display reduced frequencies of homeostatic cytokine receptors for interleukin (IL)-2 (CD25) and IL-7 (CD127), and increased expression of ICs (PD-1, CTLA-4), and cytotoxic protein granzyme B (GzmB)[53–58]. Furthermore, downregulation of co-stimulatory molecules such as CD27, and particularly CD28, are often observed in chronic viral infections and constitutes a hallmark feature of T-cell senescence and immune decline[59,60]. Using high-parameter flow cytometry, we evaluated these features in CD4+ T cells from rs6861(TT) versus rs6861(CC) genotyped HIV-1-infected individuals during the chronic stages of infection (Supplementary Fig. 3) at steady-state and upon activation with an HIV-1 peptide-pool or soluble anti-CD3 and anti-CD28.

The T-cell compartment of HIV-1-infected individuals at steady-state was characterized by a CD4:CD8 ratio <1 and frequencies of CD27-CD28-, GzmB+, and Ki-67+ cells were similar in genotyped HIV-1-infected individuals (Supplementary Fig. 6a, b). Notably, CD4+ T cells from rs6861(TT) genotyped HIV-1-infected individuals exhibited a strong reduction in the frequency of CTLA-4+ cells at steady-state as compared to the rs6861(CC) genotype (Fig. 5a, b). PD-1-expression was also decreased on proliferating (CTV-) CD4+ T cells derived from rs6861(TT) genotyped HIV-1-infected individuals (Fig. 5a, c). These

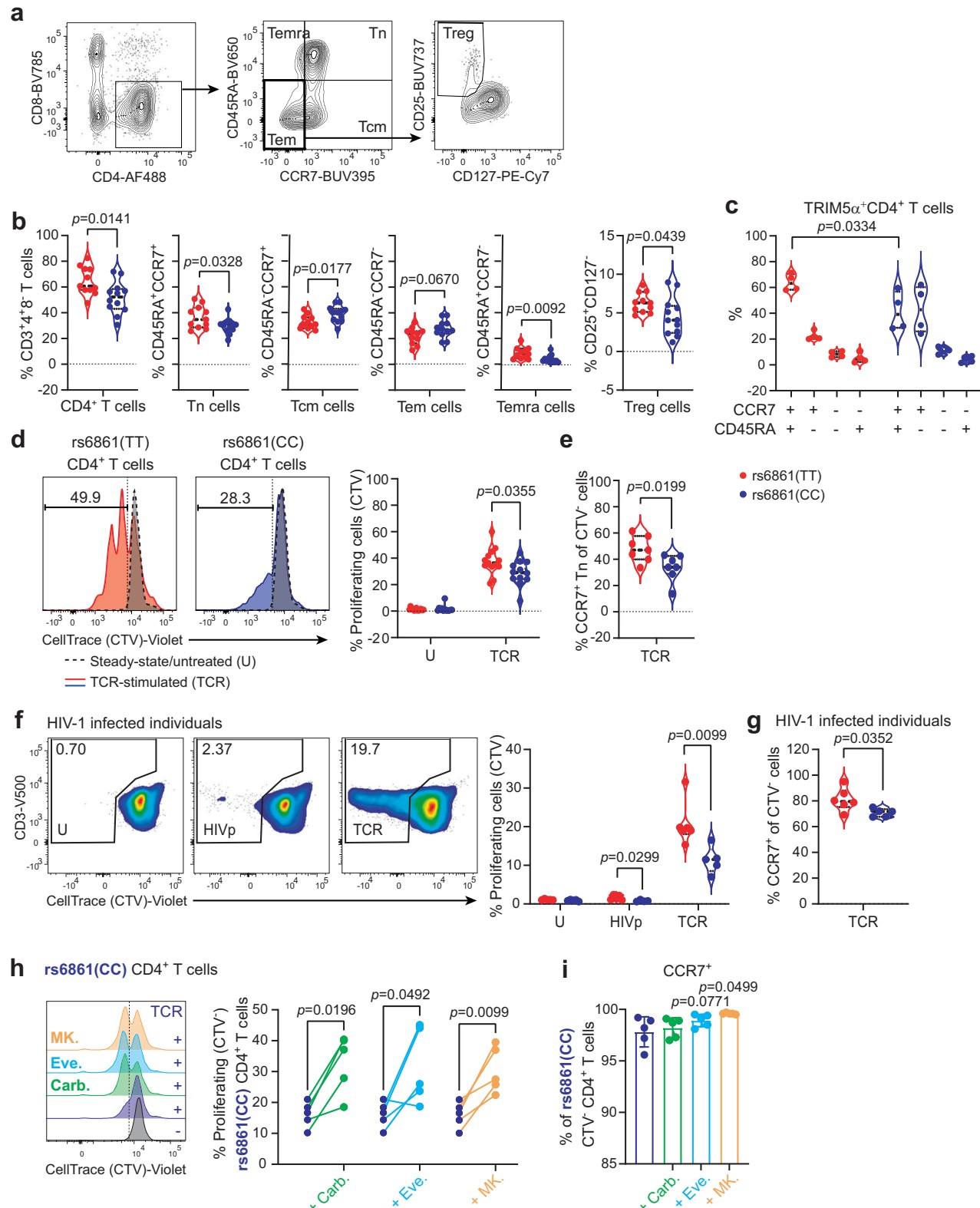

results indicate that the *ATG16L1* rs6861(TT) genotype is associated with an intrinsic reduction of the immune exhaustion signature in chronic HIV-1 infection.

Upon TCR-engagement, proliferating (CTV-) CD4+ T cells from rs6861(TT) genotyped HIV-1-infected individuals displayed increased frequencies of both CTLA-4+ and PD-1+ cells as compared to rs6861(CC), including within the proliferating CCR7+ lymph-node

homing compartment (Fig. 5d [upper panels]). Concomitantly, TCR-activated CTV- and CTV-CCR7+ rs6861(TT) genotyped CD4+ T cells displayed increased numbers of cells expressing the activation markers CD28 and increased CD25-expression (MFI) as compared to rs6861(CC) (Fig. 5d [lower panels]; the frequency of CD25+CD4+ T cells was similar between genotyped HIV-1-infected individuals [Supplementary Fig. 7]). Hence, our data suggest that following

**Fig. 4 | rs6861(TT) CD4 + T cells display augmented frequencies of T-cell subsets, enhanced proliferative capacity, and enrichment of CCR7+ cells.**
**a** Flow-cytometric representation of the gating strategy used to determine different CD4+ T cell subsets. **b** Increased frequency of total CD4+, Tn, Temra, and Treg cells and decreased frequency of Tcm cells in rs6861(TT) compared to rs6861(CC) healthy donors ($n = 12$ per genotype). **c** TRIM5α+ CD4+ T cells are predominantly restricted to the CCR7+ cell compartment and more prominent in the CCR7+ CD45RA+ Tn cell subset of rs6861(TT) individuals ($n = 4$ per genotype). **d** Increased CD4+ T-cell proliferation in response to TCR-stimulation as determined by division of CellTrace reagent over daughter cells (TT $n = 11$, CC $n = 12$), **e** particularly in the CD4+ CCR7+ Tn compartment of rs6861(TT) individuals ($n = 7$ per genotype). **f** Increased proliferation of rs6861(TT) CD4+ T cells from genotyped HIV-1-infected

individuals in response to stimulation with HIV-1 peptide pool or TCR-engagement (TT $n = 6$, CC $n = 5$), **g** particularly in the CCR7+ compartment (b-g independent two-tailed $t$-tests) (TT $n = 6$, CC $n = 5$). **h** Treatment with autophagy-enhancing pharmaceuticals carbamazepine (100 μM), everolimus (5 nM), or MK-2206 (1 μM) indicates increased CD4+ T-cell proliferation and **i** increased percentage of CCR7+ cells in rs6861(CC) donors (**h**, **i** $n = 5$ per genotype, dependent two-tailed $t$-tests, **i** mean ± SD is shown). Tn naïve, Tcm central-memory, Tem effector-memory, Temra CD45RA-expressing Tem, Treg regulatory T cells, CTV CellTrace Violet, U untreated, TCR anti-CD3/CD28 stimulation, HIV HIV-1 peptide pool stimulation, Carb. carbamazepine, Eve. everolimus, MK. MK-2206. See Supplementary Fig. 3, Supplementary Fig. 4 and Supplementary Fig. 5a, b. Source data are provided as a Source Data file.

TCR-stimulation, CD4+ T cells from rs6861(TT) genotyped HIV-1-infected individuals exhibit enhanced immune responsiveness while retaining self-limiting capacity, as evidenced by simultaneous upregulation of T-cell associated co-stimulation/activation markers and ICs, respectively. Combined, our data suggest distinct immune signatures in genotyped CD4+ T cells; rs6861(CC) CD4+ T cells displayed relatively increased exhaustion (CTLA-4+ , PD-1+ ) and immunosenescence (CD28- cell accumulation) features at steady-state whereas rs6861(TT) CD4+ T cells exhibit relatively increased proliferative capacity (CTV-), expanded CCR7+ compartment, improved T-cell activation (CD28+ , CD25+ ) and immune resolution (CTLA-4+ , PD-1+ ) upon TCR-activation (Fig. 5e).

Notably, therapeutic enhancement of autophagy in rs6861(CC) genotyped CD4+ T cells spurred the ability of these cells to exhibit the traits attributed to the rs6861(TT) genotype, supporting a relationship between increased autophagy activity and suppression of exhaustion markers CTLA-4/PD-1 and upregulation of activation marker CD25 (Fig. 5f, Supplementary Fig. 5c).

Our findings thus indicate a superior immune functionality of the CD4+ T-cell compartment of individuals carrying the *ATG16L1* SNP during chronic natural HIV-1 disease. rs6861(TT) genotyped HIV-1 infected individuals appear endowed with concomitant exceeding responsiveness and immunoregulatory capacities as evidenced by appropriate CD4+ T-cell activation and restricted IC upregulation upon TCR engagement.

## Superior CD8+ T cell responses in *ATG16L1* rs6861(TT) genotyped HIV-1-infected individuals

CD4+ T cells are essential for the induction of an antiviral CD8+ T cell response, and the decline of the CD4+ T cell compartment following HIV-1 infection is accompanied by aberrant CD8+ T-cell immunity[37,52]. Additionally, CD8+ T cells play a key role in maintaining suppression of viremia in the absence of antiretroviral therapy in individuals endowed with HIV-1 natural control[61–63]. Therefore, we next investigated the impact of the *ATG16L1* rs6861 SNP on the CD8+ T-cell compartment.

At steady state, no difference was observed in the basal autophagy levels between genotyped CD8+ T cells (Fig. 6a). However, upon TCR-engagement, rs6861(TT) CD8+ T cells displayed enhanced autophagy flux as compared to rs6861(CC) cells (Fig. 6b). Furthermore, the frequency of TRIM5α+ CD8+ T cells in response to TCR-engagement particularly increased in rs6861(TT) genotype (Supplementary Fig. 2c, d). These data suggest that rs6861(TT) CD8+ T cells display enhanced autophagy activity upon TCR-ligation.

In line with CD4+ T cells from rs6861(TT) genotyped healthy controls (Fig. 4d), rs6861(TT) CD8+ T cells demonstrated increased proliferative capacity upon TCR-mediated activation (Fig. 6c). Notably, CD8+ T cells derived from rs6861(TT) HIV-1-infected individuals (Supplementary Fig. 3) also displayed increased frequencies of proliferating cells (CTV-) in response to stimulation with an HIV-1 peptide pool and upon TCR-triggering (Fig. 6d). Furthermore, enrichment of the CCR7+ Tn compartment was evident in proliferating rs6861(TT) CD8+ T cells (Fig. 6e). Cytotoxic T-cell activation via TCR-signaling was

also analyzed by detection of CD137; a member of the tumor necrosis factor receptor (TNFR) family with co-stimulatory function, triggering pro-survival and proliferation signals in activated T cells, which can be exploited to detect antigen-specific CD8+ T cells independent of epitope specificity[64]. Notably, greater upregulation of CD137 was observed in rs6861(TT) compared to rs6861(CC) CD8+ T cells in response to TCR-engagement, suggesting that HIV-1-infected individuals carrying the *ATG16L1* rs6861(TT) genetic variant sustain a larger virus-reactive CD8+ T-cell compartment (Fig. 6f).

In exhausted CD8+ T cells, increased expression of ICs precludes hierarchical loss of effector functionality[26]. Phenotypic analysis of CD8+ T cells from genotyped HIV-1-infected individuals revealed reduced frequency of CTLA-4+ cells, but not PD-1+ cells in rs6861(TT) compared to rs6861(CC) CD8+ T cells at steady-state as well as in HIV-specific (HIV-dextramer + ) CD8+ T cells (Fig. 6g and Supplementary Fig. 8). Notably, in line with what was observed in CD4+ T cells, CD8+ T cells from rs6861(TT) genotyped HIV-1-infected individuals displayed increased co-stimulation and homeostatic signaling as determined by increased frequencies of CD28+ and CD127+ cells (Fig. 6g).

Treatment of rs6861(CC) CD8+ T cells with autophagy-enhancing drugs recapitulated the functional traits inherent to rs6861(TT) CD8+ T-cells. Carbamazepine-, everolimus-, and MK-2206-treatment all resulted in increased CD8+ T-cell proliferation compared to TCR-activated CD8+ T cells (Fig. 6h). Moreover, therapeutic autophagy enhancement in rs6861(CC) genotyped CD8+ T cells resulted in a concomitant enrichment of the CCR7+ compartment and down-regulation of exhaustion markers CTLA-4 and PD-1 (Fig. 6i and Supplementary Fig. 5d, e). Combined, these data suggest that CD8+ T cells derived from individuals carrying the *ATG16L1* rs6861(CC) genetic variant can be reprogrammed to acquire the phenotypical and functional signatures observed in rs6861(TT) T cells characterized by reduced exhaustion signatures, increased capacity for renewal of the naïve compartment, and enhanced immune response efficacy. Our findings thus highlight the potential of pharmaceutically targeting autophagy activity to boost not only CD4+ T cell (Figs. 4 and 5), but also CD8+ T-cell antiviral immunity in chronic HIV-1 infection.

## Transcriptome profiling and in-depth Th subset analysis demonstrates an augmented Treg:Th17 balance in rs6861(TT) genotyped CD4+ T cells

To gain insight into the transcriptional signatures associated with the protective effect of the *ATG16L1* rs6861 gene variant on T-cell functionality in the context of chronic viral infection, we utilized NGS RNA-sequencing technology in combination with GSEA to determine differences in the transcriptional profiles of rs6861(TT) and rs6861(CC) genotyped FACSorted CD4+ T cells at steady state or upon activation. To this end, a list of 262 genes related to T-cell inflammation/immune functionality was curated. GSEA revealed enrichment of gene networks involved in the acute response to antigenic stimuli in rs6861(TT) genotyped CD4+ T cells at steady state (Fig. 7a) as evidenced by

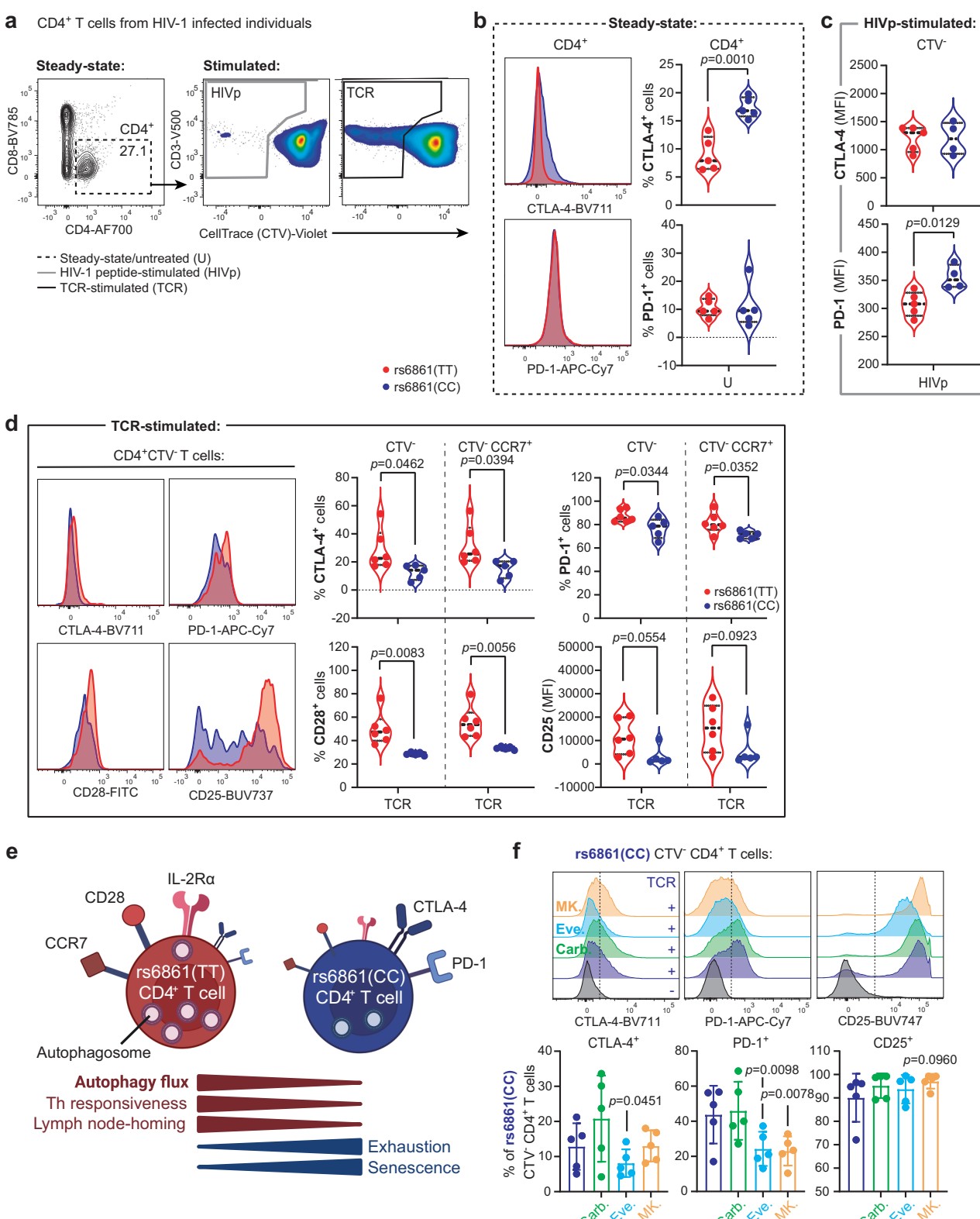

increased expression of *ICAM1* and *BTK*, which play instrumental roles in T-cell activation (Fig. 7b)[65,66]. Simultaneously, genes regulating this acute response, such as *SERPINC1* and *FCGR2B*, were upregulated in rs6861(TT) CD4+ T cells (Fig. 7a, b)[67,68]. Upon TCR-engagement (combined with IL-15 to promote memory-effector responses[24]), an enrichment of Th1-type immune response pathways, a specialized lineage of CD4+ T effector cells that promotes cell-mediated responses crucial

for host-defense against viral pathogens, was observed in rs6861(TT) versus rs6861(CC) genotyped CD4+ T cells (Fig. 7c). Strikingly, gene sets involved in the production of IL-1 family members IL-18 and IL-1β were suppressed in rs6861(TT) CD4+ T cells both at steady-state and upon TCR-engagement (Fig. 7c, d). IL-1β-production is dependent on the multi-protein inflammasome complex, of which several related genes (*NLRP2, NLRC4, CARD17,* and *CARD18*) were downregulated in

**Fig. 5 | Enhanced immune responsiveness and reduced exhaustion signatures of rs6861(TT) CD4+ T cells from HIV-1-infected individuals. a** Flow-cytometry plots showing the different CD4+ T-cell states investigated, namely at steady-state (dotted line), and either proliferating (CTV·) HIV-1 peptide pool-stimulated (gray line), or proliferating (CTV-; black line) TCR-activated CD4+ T cells. **b, c** Decreased frequency of CTLA-4+ cells at steady-state in total CD4+ T cells (*n* = 5 per genotype) (**b**) and decreased expression of PD-1 in proliferating (CTV·) CD4+ T cells upon stimulation with HIV-1 peptide pool in rs6861(TT) (*n* = 5) compared to rs6861(CC) HIV-1-infected individuals (*n* = 4) (**c**). **d** Increased frequency of CTLA-4+ , PD-1+ , CD28+, and increased CD25 MFI of CTV· CD4+ T cells, including in the CCR7+ compartment, in rs6861(TT) HIV-1 infected individuals upon TCR-engagement

(TT *n* = 6, CC *n* = 5) (**b**–**d** independent two-tailed *t*-tests). **e** Schematic representation of the phenotypical and functional characteristics of the rs6861 genotypes. **f** Pharmaceutical enhancement of autophagy flux with carbamazepine (100 μM), everolimus (5 nM), or MK-2206 (1 μM) resulted in reduced PD-1 and CTLA-4-expression and increased CD25 expression in rs6861(CC) CD4+ T cells (*n* = 5) (dependent two-tailed *t*-tests, mean ± SD is shown). CTV CellTrace Violet, U untreated, HIVp HIV-1 peptide pool stimulation, TCR anti-CD3/CD28 stimulation, MFI median fluorescence intensity, Carb. carbamazepine, Eve. everolimus, MK. MK-2206. See Supplementary Fig. 3 and Supplementary Fig. 5c. Source data are provided as a Source Data file. **e** Created with Biorender.com.

rs6861(TT) CD4+ T cells[69,70] (Fig. 7d). These data suggest that upon CD4+ T-cell activation, rs6861(TT) individuals readily upregulate anti-viral adaptive immune responses whilst downregulating pro-inflammatory pathways. This transcriptional profile further suggests the increased readiness of individuals carrying the *ATG16L1* rs6861(TT) genetic variant to respond to incoming threats while exerting greater capacity to prevent immune overactivation via self-regulatory mechanisms.

Notably, the *ATG16L1* T300A Crohn's disease variant associated with dysfunctional autophagy flux is known to impact Th1 responses as well as IL-1β-secretion[20,22,71]. Th17 cells play an instrumental role in the severity of chronic inflammatory diseases such as Crohn's disease. Additionally, Th17 differentiation, in particular Treg-to-Th17 balance, is IL-1β-driven[72–74]. Moreover, functional autophagy has been shown to control transcriptional programs in Treg cells in an mTOR-dependent manner[75]. These reports, in combination with the overrepresentation of Th1 and IL-1 transcriptional networks in rs6861(TT) genotyped CD4+ T cells observed in this study, support a differential regulation of Th subsets between genotyped individuals.

Therefore, we next performed in-depth Th subset analysis on CD4+ T cells derived from genotyped individuals. Targeted analysis of 26 common Th-associated genes indicated an increase in Treg genes (e.g. *FOXP3*, *FOXO1*, *SHMT2*) and a decrease in Th17 genes (e.g. *RORC*, *IL23R*, *IRF4*) in rs6861 (TT) versus rs6861 (CC) CD4+ T cells at steady state (Fig. 7e)[76,77]. In particular, Th17-associated *IL17F*[78] was differentially expressed between genotypes as evidenced by reduced counts per million (CPM) in TCR-stimulated rs6861(TT) CD4+ T cells, determined by single gene analysis (Fig. 7f, Supplementary Table 2) as well as quantification of intracellular IL-17F protein expression by flow-cytometry analysis (Fig. 7g). Similar frequencies of TNF-α+ , IFN-γ+ (Th1), IL-4+ (Th2), and IL-10+ (Treg) CD4+ T cells were detected between genotypes (Supplementary Fig. 10a and Fig. 7h). Notably, additional Th17 cell features, as determined by IL-17A+ and IL-22+ cell frequencies, were decreased in rs6861 (TT) CD4+ T cells (Fig. 7h). Evaluation of the canonical transcription factors associated with the main Th subsets demonstrated similar frequencies of Th1 (T-bet+ ) and Th2 (GATA3+ ) cells between genotypes (Supplementary Fig. 10b). In contrast, the CD4+ T-cell compartment from rs6861(TT) genotyped individuals displayed a concomitant reduction of Th17 (RORγt+) cell frequency and an increase in Treg (FoxP3+) cells (Fig. 7i), corroborating both the transcriptional (Fig. 7e, f) and surface phenotypical (CD25+ /CD127low; Treg; Fig. 4b) signatures associated with an immunoregulatory CD4+ T cell phenotype. Furthermore, pharmaceutical treatment of rs6861(CC) TCR-activated CD4+ T cells with autophagy enhancer MK-2206 reduced all investigated Th17-cell characteristics: the frequency of IL-17F+ , IL-17A+ , and IL-22+ CD4+ T cells, and RORγt-expression (Fig. 7j). These data show that transcriptional signatures in combination with the phenotypical and functional analyses support a higher Treg:Th17 ratio in rs6861(TT) genotyped individuals (Fig. 7k).

Taken together, the transcriptome network analysis and in-depth profiling presented here strongly indicate that T cells from rs6861(TT) genotyped individuals, while poised to mount appropriate CD4+ and

CD8+ T cell immunity towards viral infection also favor containment of Th17-mediated inflammatory processes accompanied by increased immunoregulatory T cell capacity.

## Discussion

The HIV-1 pandemic persists despite the efficacy of antiretroviral therapy in suppressing HIV-1 replication. HIV-1 has evolved mechanisms to evade and hijack host cell immune responses to establish lifelong infection. Here, we showcase host autophagy as a key regulator of protective antiviral T-cell immunity and its role in ensuing resolution of inflammatory immune responses in virologic disease progression. Our study indicates a pivotal role for autophagy in management of chronic disease in people living with HIV. By investigating the functional impact of genetic variation in the pivotal autophagy regulator ATG16L1 in chronic HIV-1 infection, we have uncovered the *ATG16L1* rs6861(TT) genetic polymorphism to be associated with improved clinical outcomes including delayed disease progression and prolonged survival rate in untreated HIV-1 infected individuals. In-depth T cell profiling analyses revealed that CD4+ and CD8+ T cells derived from HIV-1 infected individuals carrying the rs6861(TT) genotype displayed heightened autophagy activity and favorable immune functionalities including enhanced self-renewal capacity, enrichment of the naïve and lymph-node homing compartments, increased responsiveness to stimulation, and reduction of exhaustion and immunosenescence traits.

Autophagy has long been linked to T-cell functionality. Inhibition of autophagy has been shown to impair T-cell proliferation and increase IC-expression[28,39,79,80]. Enhanced autophagy-lysosomal function has been associated with extended longevity and healthy aging[81]. Furthermore, HIV-1 accessory molecules directly impact autophagy pathways in T cells. HIV-1 *Nef* was shown to block autophagy-mediated restriction[82], HIV-1 *Vpr* inhibits autophagy at the early stages of HIV-1 infection[83], and HIV-1 *Tat*-mediated hyper-activation on mTOR is required for viral production[84]. HIV-1-mediated autophagy blockage contributes to the CD4+ T cell depletion and reduced capacity to degrade cellular debris observed in HIV-1 disease[85], which has been associated with loss of T-cell homeostasis and higher rates of ill-responsive cells. Our study underpins an important role for autophagy activity in upholding T cell homeostasis, activation and survival of both CD4+ and CD8+ T cells in chronically HIV-1 infected individuals.

Markedly, *ATG16L1* rs6861(TT) genotyped individuals exhibit longer-lasting HIV-1 control by CD4+ and CD8+ T-cells as evidenced by: (i) revamped survival of the T-cell compartment with relatively higher levels of CD4+ T cells in both rs6861(TT) genotyped HIV-1-infected individuals and healthy blood donors, (ii) enrichment of naïve, antiviral Temra and regulatory T cell subsets, and (iii) increased proliferative capacity of CD4+ and CD8+ T cells, particularly CCR7+ cells, in response to both polyclonal and antigen-specific stimulation during chronic HIV-1 infection. Moreover, the defining characteristics of CD4+ and CD8+ T-cell exhaustion and immunosenescence were reduced in individuals with augmented autophagy; T cells from rs6861(TT) individuals displayed higher frequencies of IL-2 and IL-7 receptor-positive cells indicating increased response capacity to homeostatic cytokines,

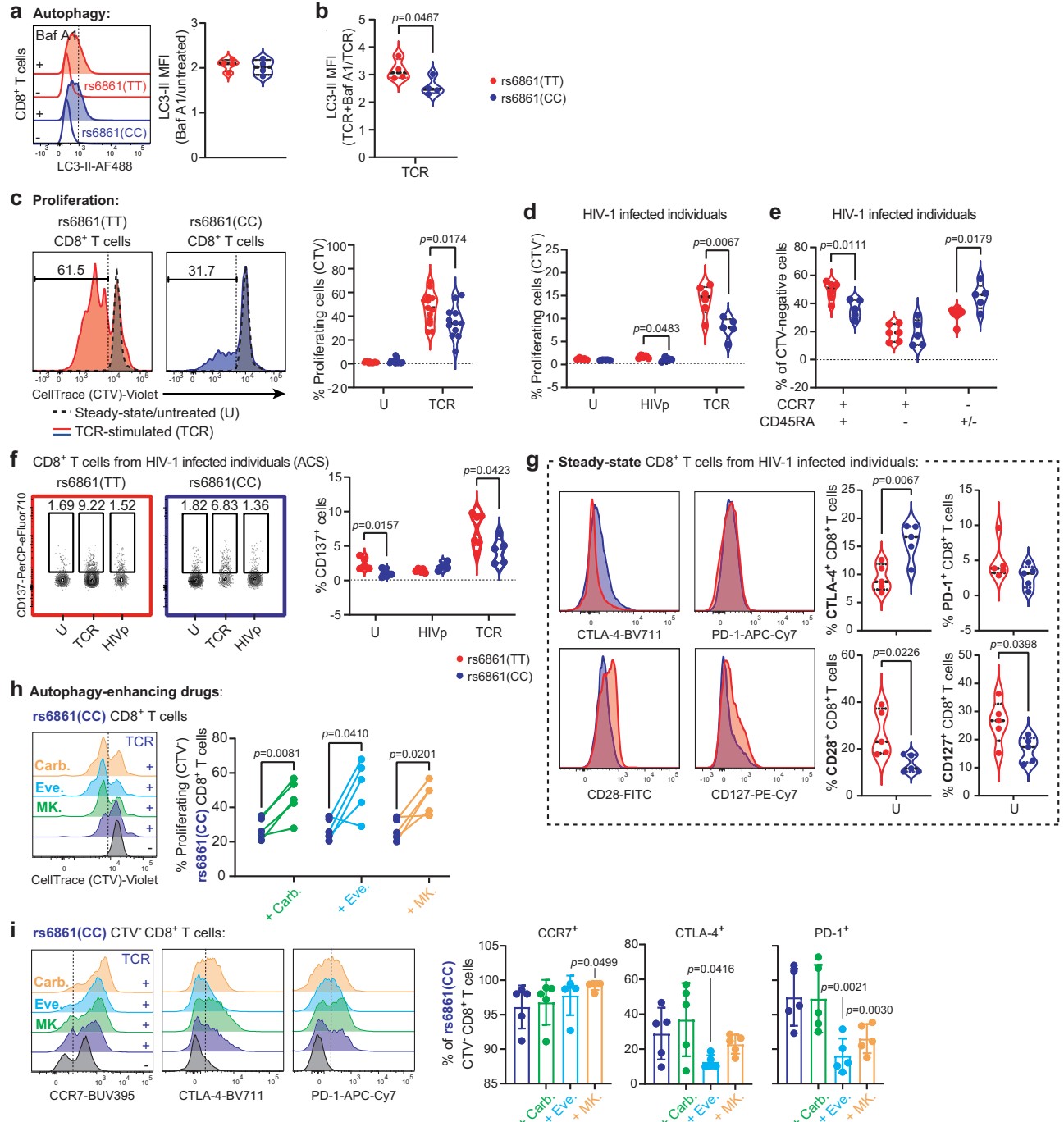

**Fig. 6 | Augmented CD8+ T cell responses preclude enhanced antiviral immunity in rs6861(TT) individuals during chronic HIV-1 infection. a, b** Flow-cytometric representation of autophagy flux as measured using intracellular LC3-II accumulation (MFI) in CD8+ T cells, showing similar baseline autophagy status between genotypes at steady-state (**a**), and increased autophagy flux upon TCR-mediated stimulation (**b**) in rs6861(TT) compared to rs6861(CC) CD8+ T cells (**a, b** n = 4 per genotype). **c** Increased CD8+ T-cell proliferation in response to TCR-stimulation (TT n = 11, CC n = 12). **d** Increased CD8+ T-cell proliferation in cells isolated from HIV-1-infected individuals upon stimulation with HIV-1 peptide pool or TCR-mediated activation. **e** Particularly the CD8+ Tn cell CCR7+ compartment of rs6861(TT) HIV-1-infected individuals demonstrated increased proliferation in response to TCR-mediated activation as compared to rs6861(CC) cells. **f** Flow-cytometric representation of CD137+ CD8+ T cells from HIV-1 infected individuals upon TCR-mediated activation, showing increased frequencies of CD137 in

rs6861(TT) compared to rs6861(CC) CD8+ T cells (d-f TT n = 6, CC n = 5). **g** Flow-cytometric representation of CD8+ T cells of HIV-1-infected individuals during chronic infection, demonstrating reduced expression of CLTA-4 but not PD-1 as well as increased expression of CD28 and CD127 at steady-state (n = 5 per genotype) (**a–g** independent two-tailed t-tests). **h** Treatment with autophagy-enhancing drugs carbamazepine (100 μM), everolimus (5 nM), or MK-2206 (1 μM) resulted in increased CD8+ T-cell proliferative capacity as compared to TCR-activation in the absence of drug treatment and **i** increased CCR7 as well as reduced frequency of CTLA-4 and PD-1-expressing CD8+ T cells isolated from rs6861(CC) individuals (**h-i** n = 5, dependent two-tailed t-tests, mean ± SD is shown). Baf A1 bafilomycin A1, MFI median fluorescence intensity, CTV CellTrace Violet, U untreated, TCR anti-CD3/CD28 stimulation, HIV HIV-1 peptide stimulation, Carb. carbamazepine, Eve. everolimus, MK. MK-2206. See Supplementary Fig. 3, Supplementary Fig. 5d, e and Fig. 8. Source data are provided as a Source Data file.

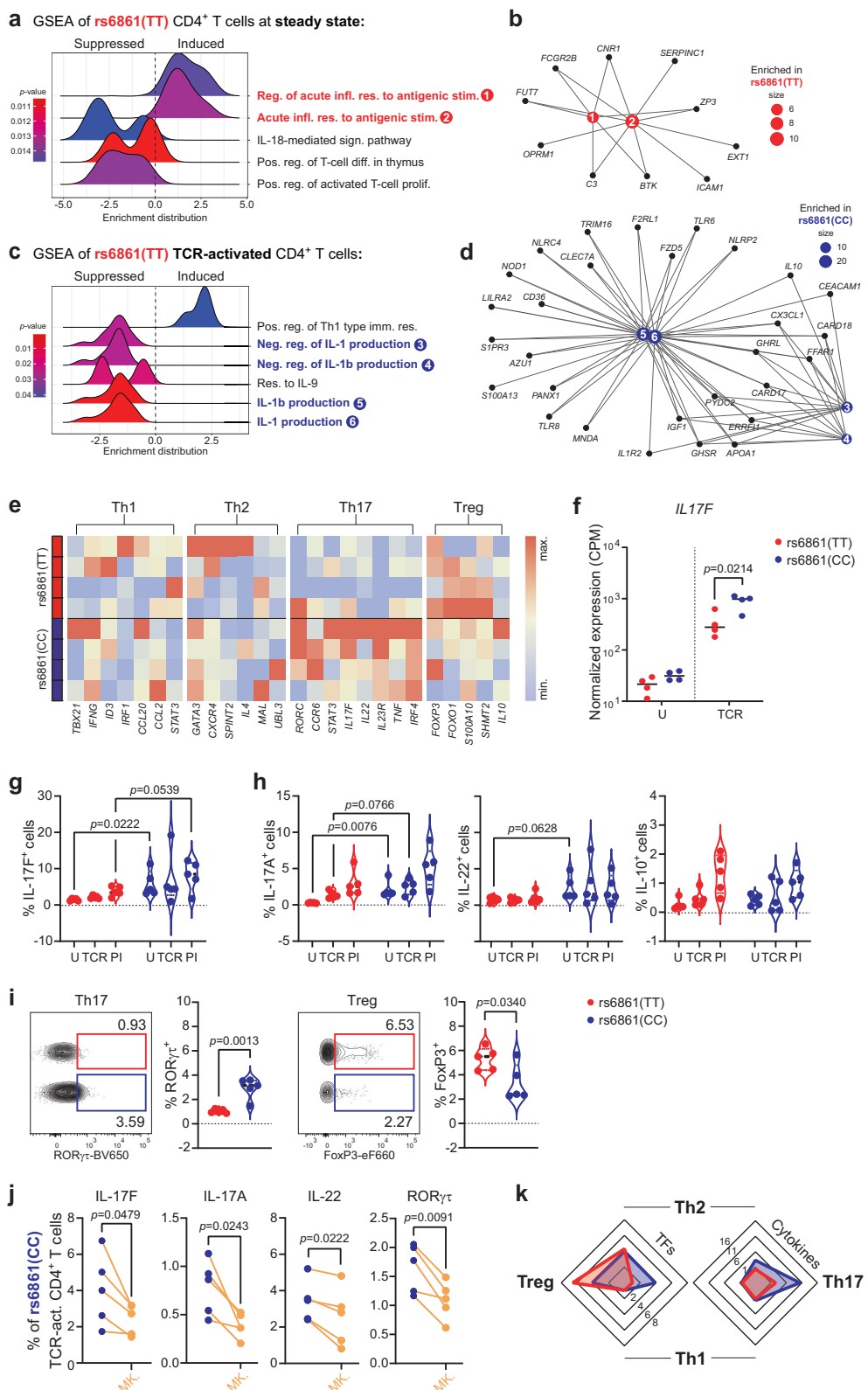

enhanced ability to upregulate co-stimulatory molecule CD28 and activation marker CD137 upon TCR-mediated stimulation as well as reduced expression of inhibitory ICs CTLA-4 and PD-1 at baseline. Notably, the accumulation of T cells deprived of CD28, a hallmark feature of chronic virologic infection and indicator of ill-responsive cells[86], is drastically reduced in rs6861(TT) genotyped individuals, strongly indicating superior T-cell activation and resistance to

repeated antigenic stimulation, warranting appropriate immune responsiveness. These findings are in concordance with a recent study showing that upregulation of CCR7 and enhanced expression of CD28 and the IL-7 receptor are distinctive properties associated with natural HIV-1 control and dependent on the autophagy-related mTOR-axis[87]. In conjunction with the restricted expression of ICs, which is being heavily investigated to reverse the deficiency of antiviral CD8+ T cells

**Fig. 7 | Transcriptomics and Th profiling identifies immune-regulatory inflammation signatures and augmented Treg:Th17 ratio in rs6861(TT) CD4+ T cells. a–d** GSEA ridgeplots and CNET plots of T-cell-related GO Biological processes visualizing expression distribution of enriched gene sets (unadjusted *p*-values) in FACSorted rs6861(TT) CD4+ T cells obtained through next-generation RNA-sequencing analyses at steady-state (untreated; **a-b**), or upon TCR/IL-15-stimulation (**c, d**), using rs6861(CC) cells as reasonable baseline. The CNET-plots depict the enriched genes associated with (regulation of) acute response to antigenic stimuli signatures in rs6861(TT) steady-state CD4+ T cells (**b**), or the enriched genes associated with (regulation) of IL-1 and IL-1β production signatures in rs6861(CC) TCR/IL15 stimulated CD4+ T cells (**d**). **e** Targeted analysis of 26 Th-associated genes at steady-state indicates upregulated transcription of Treg-related genes and downregulation of Th17-related genes in rs6861(TT) CD4+ T cells at steady-state. **f** Single gene expression analysis (CPM) of *IL17F* (**a–f** *n* = 4 per genotype) upon TCR + IL-7-stimulation **g** and intracellular flow-cytometric analysis of IL-17F-protein expression indicates reduced IL-17F levels in rs6861(TT) compared to rs6861(CC) CD4+ T cells. **h** Frequency of Th17 cytokine-producing IL-17A+ and IL-22+ cells are reduced in rs6861(TT) compared to rs6861(CC) CD4+ T cells,

while IL-10+ cell frequency is similar between genotypes as measured by intracellular flow cytometry. **i** Intracellular flow cytometry showing that Th17-transcription factor RORγt is reduced and Treg-transcription factor FoxP3 is increased in rs6861(TT) compared to rs6861(CC) CD4+ T cells at steady-state (**f–i** independent two-tailed *t*-tests). **j** Pharmacological treatment of rs6861(CC) CD4+ T cells with autophagy-enhancing compound MK-2206 (1 μM) reduces Th17-cell characteristics (dependent two-tailed *t*-tests). **k** Spider graphs displaying distribution of Th profiles according to transcription factors expression and cytokines produced by rs6861(TT) compared to rs6861(CC) CD4+ T cells at steady-state displaying an augmented Treg:Th17 ratio in rs6861(TT) genotyped individuals (**g–k**: *n* = 5 per genotype). Reg. regulation, infl. inflammation, res. response, stim. stimulus, sign. signaling, pos. positive, diff. differentiation, prolif. proliferation, neg. negative, Imm. immune, CPM counts per million, U steady-state, untreated, TCR anti-CD3/CD28, act. activated, PI PMA/Ionomycin, TFs Transcription factors, MK. MK-2206. See Supplementary Tables 2 and 3, and Supplementary Fig. 9. Source data are provided as a Source Data file. RNA-seq. data are accessible through GEO Series accession number GSE253769.

to adequately control HIV-1 infection and eliminate latently infected cells, our data underscore the pivotal role of functional T-cell autophagy to rejuvenate CD4+ and CD8+ T cellular responses, contribute to the recovery of the T-cell compartment by inducing naïve T-cell generation and revamp immune responsiveness in chronic HIV-1 infection. Further investigation using HIV-1 treatment cohorts will give insight into the impact of this *ATG16L1* SNP on immune reconstitution profiles and viral reservoir size in people living with HIV.

Besides CD4+ depletion and T-cell dysfunction, persistent inflammation is another seminal feature of HIV-1-associated immune dysregulation that lingers in life-long cART-treated HIV-1-infected individuals and represents a risk factor for people living with HIV to develop severe comorbidities including neurodegenerative and cardiovascular diseases[3,58]. Notably, our study shows that *ATG16L1* rs6861(TT) genotyped individuals exhibit a superior capacity to curb IL-1β-mediated proinflammatory networks accompanied by tipping the Th17:Treg balance towards an immunoregulatory phenotype. Next-generation mRNA sequencing uncovered a unique capacity for T cells derived from rs6861(TT) genotyped individuals to fine-tune pro-inflammatory pathways, in particular IL-1β-mediated signaling. Selective suppression of inflammasome-associated genes (such as *NLRP2, NLRC4, CARD17* and *CARD18*) and IL-1β-inducing gene networks (such as *TLR6, TLR8* and *TRIM16*) upon TCR-mediated T-cell activation in conjunction with positive regulation of autophagy flux (*NBR1, LIPA, MANBA, ABL1*) suggest that rs6861(TT) genotyped individuals with intrinsically-enhanced degradative autophagy capacity are poised to subdue the release of pro-inflammatory mediators and pyroptosis-driven CD4+ T-cell death. T-helper profiling of rs6861(TT) genotyped T cells demonstrated an enrichment of immunoregulatory gene signatures (such as *FOXP3* and *FOXO1*) accompanied by concomitant increase of T regulatory markers (FoxP3-expression and CD25+ CD127-surface phenotype) and decreased Th17-associated features (including *RORC, CCR6, IL23R*, RORγt-expression, and IL-17A/F-production). The increased Treg/Th17 ratio observed in rs6861(TT) genotyped donors is in line with several renowned studies reporting the role of mTOR-dependent autophagy and inflammasome-mediated IL-1β-activation in driving Treg and Th17 polarization, respectively[75,88–90]. A stable Treg:Th17 ratio is imperative to curb excessive inflammatory responses. The specific enrichment of immunoregulatory networks and the inflated Treg:Th17 ratio in rs6861(TT) genotyped individuals as well as the suppression of inflammatory Th17 features upon treatment with autophagy-enhancing drugs highlight a major role for heightened host autophagy flux to provide a superior capacity to manage pro-inflammatory processes, thereby limiting persistent immune activation associated with favorable clinical outcomes in chronic HIV-1 infection.

Therapeutic exploitation of autophagy has advanced in parallel to the investigation of its diverse roles in human diseases. Various small-molecule modulators directly or indirectly targeting key components of autophagy machinery have been developed[91–94]. These autophagy-modulating compounds have shown promising results in preclinical models and many have already been FDA-approved for treatment of i.e. epilepsy, cancers, and transplantation[95]. Recently, we have shown that treatment of human mucosal tissues with clinically-approved autophagy-enhancing drugs prevents HIV-1 acquisition and suppresses ongoing HIV-1 replication by CD4+ T cells[46]. Notably, in the current study we demonstrated that treatment with autophagy-enhancing pharmaceuticals reprogrammed CD4+ and CD8+ T-cells from rs6861(CC) genotyped individuals to acquire increased T-cell immune response efficacy tributary to the *ATG16L1* rs6861(TT) carriers as evidenced by enhanced proliferative capacity and renewal of the CCR7+ compartment, downregulation of exhaustion signatures, and reduction of Th17 polarization. Taking into account the notorious roles for proficient autophagy in promoting lysosomal viral degradation and T cell homeostasis[5,79] as well as the therapeutic potential of autophagy-enhancing drugs to advance the magnitude and regulation of CD4+ and CD8+ T-cell immune responses, it is plausible to propose that intrinsically heightened levels of autophagy activity in the T cell compartment may contribute to the increased containment of chronic HIV-1 disease in rs6861(TT) genotyped individuals. Follow-up studies utilizing specific genome editing and identical genetic backgrounds are required to establish causal relationships between SNP-causing phenotypes as the function of the rs6861 genetic variant is yet unknown. Building upon the reports describing the importance of autophagy on the natural control of HIV-1 infection in long-term nonprogressors and elite controllers[10,11], this study focusing on genetic variation in the autophagy pathway in chronically HIV-1 infected individuals as well as therapeutic enhancement of autophagy ex vivo further underscore the relevancy of autophagy in controlling T cell immunity.

Approaches targeting regulatory autophagy mechanisms or gene editing strategies directed at core autophagy genes are anticipated to expand the discovery of autophagy-selective pharmacological agents. Our findings highlight host autophagy, and in particular ATG16L1, as an emerging target for HIV-1 therapies, underscoring the relevancy of repurposing clinically-approved autophagy drugs to restore immune function in chronic HIV-1 infection. Moreover, our study focusing on the hitherto unidentified autophagy-enhancing *ATG16L1* rs6861 polymorphism provides a relevant model to investigate the in vivo impact of heightened autophagy in the severity of HIV-1-associated comorbidities as well as other persistent viral infections, inflammatory disorders and diseases characterized by autophagy dysfunction[96].

Next-generation host-directed therapies targeting autophagy are thereby a promising avenue to boost antiviral CD4+ and CD8+ T cell responses and dampen exacerbated inflammation to advance management of chronic disease in people living with HIV.

## Study limitations
This study was limited by the cohort size, availability and quality of cryopreserved samples (frozen between 1983–1990) of rs6861(TT) genotyped PBMC obtained from untreated HIV-1-infected individuals. Additionally, limited cell numbers prevented more extensive characterization of phenotypical and functional differences in genotyped HIV-1-infected individuals other than presented here. Results indicating modest differences that failed to reach statistical significance may also have been affected by sample size of PBMCs derived from untreated HIV-1 MSM. Furthermore, lack of replication of the clinical impact of *ATG16L1* genotype identified within ACS cohort in other available cohorts of HIV-1 disease progression might represent a potential weakness of the study. However, it is important to note that these available cohort studies perform analyses in extreme patient groups with pre-determined clinical characteristics, and are thereby not comparable to the ACS cohort analyses encompassing the complete follow-up data from the entire seroconverter cohort irrespective of clinical symptoms and progression phenotypes. Of note, in our manuscript we build up on the cohort genetic and clinical data with in-depth functional analyses in PBMCs derived from chronically HIV-1 infected individuals. The immune phenotyping and immune functionality analyses ex vivo thereby substantiate the association of the *ATG16L1* genotype with phenotypically improved clinical disease outcomes and superior CD4+ and CD8+ T cell immunity in chronic HIV-1 infection. Molecular characterization of the SNP was also beyond the scope of this manuscript. Follow-up studies exploring how the rs6861 SNP alters the biology of ATG16L1 will inform on the causal link between the occurrence of this SNP and its reported functional implications as well as further support the advancement of the design of selective autophagy-targeting therapies for human viral diseases and other immune-related disorders associated with dysfunctional autophagy.

## Methods
### Study approval
This study has been conducted in accordance with the ethical principles set out in the declaration of Helsinki. The study was approved by the Amsterdam UMC institutional Medical Ethics Review Committee of the University of Amsterdam (Amsterdam, the Netherlands), the Ethics Advisory Body of the Sanquin Blood Supply Foundation (Amsterdam, the Netherlands), and the board of the Amsterdam Cohort Studies (ACS) on HIV infection and AIDS (Amsterdam, the Netherlands). Written consent was obtained from ACS study participants. Use of buffy coats from healthy blood donors is not subjected to informed consent according to the Medical Research Involving Human Subjects Act and the Medical Ethics Review Committee of the Amsterdam UMC. All methods were performed in accordance with the relevant guidelines and regulations as stated in the Amsterdam UMC Research Code.

### Study population
The Amsterdam Cohort Studies on HIV infection and AIDS is a prospective study among men who have sex with men (MSM) that started in 1984[97]. The HIV-1-infected MSM were enrolled in the cohort between October 1984 and March 1986, and actively followed-up until May 1996 and none of the 304 genotyped MSM received effective ART before AIDS diagnosis as this was not available at the time. Eigenstraat was employed to explicitly model ancestry differences among participants (implemented in Eigensoft) and was used for the identification of outliers in the HIV-1 infected population who were subsequently removed from analysis as described in van Manen et al.[31]. Complete

follow-up data during untreated course of infection of 304 genotyped MSM with known date of seroconversion, irrespective of disease course profile, including CD4 cell count in blood and set-point plasma viral load, was available for survival analyses[31]. Viral load set-point was defined as the relatively steady level of HIV-1 RNA at 18–24 months after seroconversion[31,98]. Viral load in the serum was determined using nucleic acid sequence-based amplification assays as a measure of viral replication. From these, 6 homozygous minor (TT) and 6 homozygous major (CC) for the rs6861 *ATG16L1* SNP were selected for phenotypical and functional flow cytometry analyses; cryopreserved PBMCs from these individuals were selected between >24 months after seroconversion and >12 months prior to AIDS diagnosis (Supplementary Fig. 3). Individuals carrying confounding SNPs were excluded. Confounding SNPs were considered to be the following: CCR5Δ32, HLA-B27, HLA-B57, and homozygous minor for *ATG16L1* T300A[20,32–35], as the first three genotypes have previously been associated with HIV disease progression and the latter with autophagy dysregulation. PBMCs from 1 homozygous major (CC) HIV-1-infected individual were excluded from flow cytometric analyses due to low cell viability. Additionally, PBMCs isolated from buffy coats derived from blood from healthy individuals supplied for research purposes by Sanquin Blood Supply Foundation (Amsterdam) were included in this study.

### PBMC isolation
PBMCs were isolated from buffy coats derived from blood of healthy donors (Sanquin Blood Supply Foundation) via density gradient centrifugation and subsequently frozen until use. Blood donations are anonymized without access to information (such as age, sex, relatedness) pertaining the individual donors[99]. In brief, diluted blood was layered on top of Lymphoprep (Axis-Shield) and centrifuged for 22 min at room temperature and 1000 g with no break after which the mononuclear cell fraction was aspirated, washed and frozen in buffer containing 10% dimethyl sulfoxide (DMSO; Amresco) and minimum 30% fetal bovine serum (FBS; Biological Industries) at −196 °C. Frozen PBMC from HIV-1-infected individuals (ACS) and healthy donors (Sanquin Blood Supply Foundation) were thawed in buffer containing minimum 30% FBS and no antibiotics, washed and left to recover in the incubator (37 °C and 5% CO2) in Roswell Park Memorial Institute (RPMI; Sigma-Aldrich) medium containing 10% FBS and 1% penicillin/streptomycin (P/S; Invitrogen), 'T-cell medium' used for all assays unless described otherwise, for a minimum of 16 h to recover. Automated cell counting was used to determine the number of viable cells.

### Genotyping
*ATG16L1* genotypes T300A rs2241880 and rs6861 (See Fig. 1a) on blood samples of healthy donors (Sanquin Blood Supply Foundation) were determined using the TaqMan Sample-to-SNP kit (ThermoFisher) according to the manufacturer's instructions. Relevant genotypes of HIV-1-infected individuals was determined by the ACS.

### Cellular staining and flow cytometry
For surface-molecule staining, cells were incubated with a cocktail of titrated antibodies (see Supplementary Table 3 for a full list of antibodies) and LIVE/DEAD Fixable viability dye (ThermoFisher) diluted in PBS for 30 min, shielded from light, shaking at 600 strokes/min at 4 °C, washed twice, then fixated using 4% PFA (Electron Microscopy Sciences), washed and stored in PBS. HIV-specific CD8+ T cells were detected using MHC Dextramer technology according to the manufacturer's instructions (Immudex); MHC Class I HIV GAG HLA-A*0201 (SLYNTVATL) and HIV GAG HLA-B*0702 (GPGHKARVL) based on human leukocyte antigen (HLA)-typing performed by the ACS, which indicated most selected HIV-1-infected individuals to be HLA-A2 ($n = 6$) or HLA-B7 ($n = 6$) positive. For intracellular staining (except for LC3-II, see next paragraph), surface-stained cells were washed and fixated using Fixation/Permeabilization reagent (FoxP3/Transcription

Staining Buffer Set; eBioscience) for 15 min, shielded from light, shaking at 600 strokes/min at 4 °C, washed once, then stained intracellularly with a cocktail of titrated antibodies diluted in Permeabilization Buffer (FoxP3/Transcription Staining Buffer Set; eBioscience) for 30 min, shielded from light, shaking at 600 strokes/min at 4 °C, then washed twice and stored in PBS. Samples were acquired within 24 h after staining on a Fortessa Cell Analyzer (Becton Dickinson) using FACSDIVA software (version 10.8.1). Ultracomp eBeads (eBioscience) were used to determine spectral overlap. Data files were analyzed using FlowJo software (version 10.8.1). Debris and dead cells and were excluded based on forward- and sideward-scatter and LIVE/DEAD fixable viability dyes. Subsequently, CD3+CD4+CD8- (CD4+ T cells) and CD3+CD4-CD8+ (CD8+ T cells) were gated. Details on extended gating of T-cell subsets are provided in the figures where appropriate.

## Autophagy flux

To allow LC3-II accumulation, PBMCs were treated with lysosome-targeting Bafilomycin A1 (InvivoGen) at 50 nM for 4 h or left untreated[36]. To determine autophagy flux upon TCR-mediated activation, PBMCs were first activated for 16 h with soluble 2 µg/mL anti-CD3 and 2 µg/mL anti-CD28 (both Sanquin) followed by incubation with 50 nM Bafilomycin A1 for 4 h. Anti-LC3 monoclonal antibody was conjugated to AF488 using the Lightning-Link® Rapid Antibody Labeling kit (Expedeon) according to the manufacturer's instructions. Viable cells were suspended in 0.05% saponin (Sigma-Aldrich) for 15 min at room temperature to flush out LC3-I[36]. Cells were washed twice with PBS, then stained with the conjugated LC3-AF488 in combination with antibodies against surface receptors in 0.05% saponine buffer for 30 min, shielded from light, shaking at 600 strokes/min at 4 °C. After incubation, cells were washed twice with PBS, then fixated using 4% PFA, washed, stored in PBS and acquired on a Fortessa Cell Analyzer.

## T-cell stimulation

PBMCs were TCR-stimulated for 16 h with soluble 2 µg/mL anti-CD3 and 2 µg/mL anti-CD28 (both Sanquin), 10 ng/mL phorbol 12-myristate 13-acetate (PMA) and 1 µg/mL ionomycin (both Sigma-Aldrich), or left untreated. To measure intracellular cytokines, 10 µg/mL brefeldine A (Sigma-Aldrich) was added 1 h after addition of anti-CD3/CD28 or PMA/ Ionomycin. After incubation and cellular staining, cells were acquired on a Fortessa Cell Analyzer.

## T-cell proliferation and immunophenotyping

To determine proliferation capacity and phenotype, PBMCs were first labeled with CellTrace™ Violet Cell Proliferation kit (ThermoFisher) according to the manufacturer's instructions. Cells were then activated using either 0.2 µg/mL anti-CD3 and 2 µg/mL anti-CD28 or 4 µg/mL (2 µg/mL/peptide) HIV-1 peptide pool or left untreated for 4 days. HIV-1 peptide pools against HIV-1 subtype B (consensus) Gag and Nef regions were obtained through the NIH HIV Reagent Program, Division of AIDS (NIAID, NIH). After incubation and cellular staining, cells were acquired on a Fortessa Cell Analyzer.

## Autophagy flux reporter cell line

Autophagy flux reporter cells were generated via retroviral transduction of U87.CD4.CCR5 cells (obtained via the NIH AIDS Reagent Program, Division of AIDS, NIAID, NIH) cells with pBABE-mCherry-GFP-LC3 (Addgene 22418) as described previously[100] referred to herein as U87.LC3-mCherry-GFP cells. Cells were maintained in Iscoves Modified Dulbecco's Medium (IMDM, Thermo Fisher Scientific) supplemented with 10% FBS and 1% penicillin/streptomycin. To confirm autophagy-flux-enhancing potential of therapeutics, U87.LC3-mCherry-GFP cells were treated for 24 h with 100 uM carbamazepine (Tocris), 3 nM everolimus, or 5 uM MK-2206 (both Cayman Chemical) after which GFP reduction was determined on a Fortessa Cell Analyzer.

## Treatment with autophagy-enhancing drugs

Non-toxicity of the selected autophagy-enhancing therapeutics on CD4+ and CD8+ T cells was confirmed in the range 50–200 uM carbamazepine, 2.5–10 nM everolimus, and 0.5–2 uM MK-2206. To evaluate the impact of autophagy-enhancing drugs on immune phenotype, PBMCs were activated for 16 h with soluble 2 µg/mL anti-CD3 and 2 µg/mL anti-CD28 in combination with 100 µM Carbamazepine, 5 nM Everolimus, or 1 µM MK-2206. To determine the impact of autophagy-enhancing drugs on transcription factor or cytokine expression, PBMCs were first labeled with CellTrace™ Violet Cell Proliferation kit and then stimulated for 4 days with 0.2 µg/mL anti-CD3 and 2 µg/mL anti-CD28 in combination with 100 µM Carbamazepine, 5 nM Everolimus, or 1 µM MK-2206. After incubation and cellular staining, cells were acquired on a Fortessa Cell Analyzer.

## Fluorescence-activated cell sorting (FACS) and T-cell activation for RNA-sequencing

Four homozygous minor (TT) and four homozygous major (CC) for rs6861 ATG16L1 SNP healthy individuals (Sanquin) were selected. Individuals homozygous major for the confounding ATG16L1 T300A SNP rs2241880 were excluded[20]. PBMCs from selected individuals were surface-stained with anti-human CD3-APC-FIRE750, CD4-AF488, and CD8-PerCP-Cy5.5 (see Supplementary Table 3) and resuspended in PBS containing 1% FBS. Debris and dead cells and were excluded based on forward- and sideward-scatter and CD4+ T cells (CD3+CD4+CD8-) were sorted using a FacsAria Cell Sorter (Becton Dickinson). After sorting, cells were pelleted and plated at 250,000 cells/condition in 100 µL T-cell medium in 96-well round-bottom plates. Subsequently, cells were activated for 16 h with 0.06 µg/mL anti-CD3, 1 µg/mL anti-CD28 incombination with 2 ng/mL IL-7 or 2 ng/mL IL-15 (both PeproTech)[24] or left untreated.

## RNA isolation

After FACSorting and subsequent activation (see previous paragraph), cells were pelleted, washed once with PBS, and suspended in 700 uL buffer/sample, vortexed for 15 s, and immediately frozen at −80 °C until RNA isolation according to the miRNAeasy Micro Kit (Qiagen) manufacturer's instructions, which was utilized for RNA isolation. Frozen isolated RNA was transported to Qiagen for sequencing.

## Next generation RNA-sequencing

Quality assurance and next generation sequencing of RNA samples were outsourced to Qiagen for quality control and UPX 3' Transcriptome Sequencing according to their in-house methods. All samples passed quality control as determined by sufficient RIN scores and sufficient number of reads (>19,000). Library preparation was done using the QIAseq UPX 3' Transcriptome Kit. A total of 10 ng purified RNA/sample was converted into cDNA NGS libraries. During reverse transcription, each sample was tagged with a unique ID, and each RNA molecule was tagged with a unique molecular index (UMI). The samples were then pooled into one tube for downstream processing. Quality control of the library preparation was performed using capillary electrophoresis (Agilent DNA 7500 Chip) and the libraries were quantified using qPCR. Based on quality of the inserts and the concentration measurements, the libraries were pooled in equimolar ratios. The library pool was sequenced on a NextSeq®500 sequencing instrument according to the manufacturer instructions with 100 bp read length for read 1 and 27 bp for read 2. Raw data was demultiplexed and FASTQ files for each pool were generated using the bcl2fastq Conversion Software (Illumina® Inc.). The "Demultiplex QIAseq UPX 3' reads" tool of Qiagen's CLC Genomics Workbench 21.0.1 was used to demultiplex the raw sequencing reads according to the sample indices. The "Quantify QIAseq UPX 3' workflow" was used to process the demultiplexed sequencing reads with default settings. The reads were then mapped to the human genome GRCh38 and annotated

using the NCBI RefSeq GRCh38.p12 annotation. In short, reads were annotated with their UMI and then trimmed for poly(A) and adapter sequences, minimum read lengths (15 nucleotides), read quality, and ambiguous nucleotides (maximum of 2). They were then deduplicated using their UMI and mapped to the Human genome GRCh38. On average, >5 million reads were sequenced and assigned per sample. The sequencing quality, measured with Phred values, was on average on a high level over the whole read length for all samples. Mapping rates to the genome were high, ranging from 75–90% and the majority of reads mapped to genes (-80%) annotated as protein coding. Principle component analysis (PCA; performed with ClustVis)[101] revealed clear separation of steady-state and TCR-triggered CD4 + T cells; count per million of the 500 genes with the highest variance were used to plot the PCA with unit variance scaling (Supplementary Fig. 9).

### Differential gene expression (DEG) and gene-set enrichment analysis (GSEA)

DEG analysis was performed on RNA-sequencing data using the "Empirical analysis of DGE algorithm" of Qiagen's CLC Genomics Workbench 21.0.1 with default settings. This is an implementation of the "Exact Test"[102] for two-group comparisons and incorporated into EdgeR; a Bioconductor software package[103] using an over-dispersed Poisson model to account for biological and technical variability and empirical Bayes methods to moderate the degree of overdispersion across transcripts and suitable even for minimal levels of replication. GSEA[104] (Kolmogorov–Smirnov test) was performed using the R package clusterProfiler[105] with the following parameters: only genes actively transcribed in our samples were used as background gene set (-21,000 to 22,000 per comparison), minimum gene-set size = 5 genes and maximum geneset size = 500 genes with a cutoff p-value = 0.05. Cells derived from rs6861(CC) genotype individuals were used as baseline. Heatmaps were created using open-source MORPHEUS software (Broad Institute, https://software.broadinstitute.org/morpheus) without clustering. To determine basal differences in autophagy-related genes between rs6861(TT) and rs6861(CC) genotyped CD4 + T cells, steady-state RNA-sequencing data (obtained as described above) and the gene toolbox curated by Bordi et al.[42] were employed. The Bordi toolbox consists of 604 annotated autophagy-related genes subdivided into 6 main categories: (i) mTOR and upstream pathways, (ii) autophagy core, (iii) autophagy regulators, (iv) mitophagy, (v) docking and fusion and (vi) lysosome and lysosome-related genes. To evaluate differences in T-cell inflammation/immune functionality between genotypes at steady-state and upon TCR-engagement, a customized list of 262 T-cell inflammation/immune-related genes was created by mining the GO Biological Process gene set library (c5.go.bp.v7.4)[106] using the terms: T cell, T helper, Interleukin, Interferon gamma, Chemokine, Inflammasome, and Antigen. Additionally, a set of 26 genes highly associated with T-helper signatures was curated by mining existing literature. Single-gene expression of counts per million was visualized using GraphPad Prism.

### Statistical analyses

Data collection was performed in Microsoft Office Excel 2016. Kaplan–Meier and Cox proportional-hazard analyses were performed to study the relation between SNPs in *ATG16L1* and disease progression. The following endpoints were considered for analysis: AIDS-defining events according to the 1987 Center for Disease Control (CDC) definition, AIDS-defining events including CD4 + T-cell counts below 200 cells/μL according to the 1993 CDC definition and AIDS-related death. Individuals who started effective cART or who were lost to follow-up were censored. Differences in in vitro HIV-1 viral replication were analyzed by two-tailed t-test. Bonferroni correction for multiple testing was used where indicated. Statistical significance of differences in flow cytometry data was tested using independent

(unpaired samples) or dependent (paired samples) two-tailed t-tests as specified in the legends; the software package GraphPad Prism (version 9) was used to analyze data and perform statistical analyses and data are shown as violin plots depicting median and interquartile ranges or as bar graphs depicting the mean and standard deviation. Individual symbols are representative of distinct samples. Differential gene expression data was performed using both adjusted and unadjusted p-values depending on the context of the analysis (as indicated in the legends) and were corrected for multiple testing using the FDR/Benjamini-Hochberg method. Unadjusted p-values were used for GSEA (Kolmogorov–Smirnov) analyses. Exact p-values are shown in the figures.

### Reporting summary

Further information on research design is available in the Nature Portfolio Reporting Summary linked to this article.

## Data availability

The RNA seq data discussed in this publication have been deposited in NCBI's Gene Expression Omnibus[107] and are accessible through GEO Series accession number GSE253769. This article does not contain any original code. Requests for data should be made to and will be fulfilled by C.M.S. Ribeiro (c.m.ribeiro@amsterdamumc.nl), provided the data will be used within the scope of the originally provided informed consent. Source data are provided with this paper.

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

## Acknowledgements

The authors would like to thank all individuals participating in the Amsterdam Cohort Studies for their contribution. The Amsterdam Cohort Studies on HIV infection and AIDS, a collaboration between the Public Health Service Amsterdam, the Amsterdam University Medical Center (UMC) of the University of Amsterdam, Medical Center Jan van Goyen, and the HIV Focus Center of the DC-Clinics, which are part of the Netherlands HIV Monitoring Foundation and are financially supported by the Center for Infectious Disease Control of the Netherlands National Institute for Public Health and the Environment. We also thank all volunteers donating blood for scientific research at Sanquin Blood Supply Foundation for their contribution and access to study material; the NIH HIV Reagent Program for providing reagents; the Cellular Imaging Core Facility of the Amsterdam UMC for assistance during flow-cytometric data acquisition; and QIAGEN for providing sequencing services. This study was funded by the Dutch Research Council (NWO) VENI (863.13.025, CMSR) and NWO-ZonMw VIDI (91718331, CMSR) and ASPASIA (015.014.030, CMSR) grants. Graphical illustrations in Fig. 1b, e and Fig. 5e were created with BioRender.com (https://biorender.com) under academic subscription.

## Author contributions

R.R.C.E.S., N.A.K. and C.M.S.R. designed the experiments; T.B., B.B-N. and N.A.K. performed and analyzed the GWA scan data; A.K. analyzed the RNA-seq. data; R.R.C.E.S., B.B-N., A.P.M., K.P. and A.G. prepared or collected genotyped PBMC samples for T cell profiling; A.P.M. and K.P. contributed to the acquisition of data using autophagy-modulating drugs. R.R.C.E.S. performed the T-cell profiling, autophagy experiments and data analyses; R.R.C.E.S., N.A.K. and C.M.S.R. contributed to the data interpretation; R.R.C.E.S. and C.M.S.R. wrote the initial and revised versions of the manuscript with input from all authors; and C.M.S.R. acquired funding and supervised all the aspects of the study.

## Competing interests

The authors declare no competing interests.
