## [Peer Review File · Nature Communications]

Autophagy-enhancing ATG16L1 polymorphism is associated with improved clinical outcome and T-cell immunity in chronic HIV-1 infectionREVIEWER COMMENTS

Reviewer #1 (Remarks to the Author):

This manuscript describes the effect of a specific polymorphism in ATG16L1 (rs6861) on autophagy and its role in improving T-cell immunity in persons chronically infected with HIV-1. The investigators convincingly show that the rs6861 SNP functionally enhances autophagy and improves clinical outcomes including AIDS diagnosis and survival of those chronically infected with HIV-1. The work expands on the importance of autophagy in HIV-1 infection that had been largely unappreciated until the past decade.

Specific Comments

1. Figure 1 shows that there is a slight increase in the viral set-point and an increase in CD4 count for those with the Minor genotype. What are the intracellular levels of HIV-1 in the different genotypes?
2. Presumably, when some participants were diagnosed with HIV-1, they were started on ART and have survived. For those that are fully suppressed, what is the reservoir size of the Minor genotypes versus Heterozygous and Major genotypes? It would be of considerable interest if those with the Minor genotype have a smaller reservoir size.
3. In figure 2c, engagement of TCR using soluble CD3 and anti-CD28 results in increased autophagy in the Minor genotype. It would be of interest to determine if HIV-1 GU-rich ssRNA itself shows a similar increase in autophagy in the Minor group.
4. Figure 2e. Other than a single outlier in the Major genotype group, there appears to be very little difference in the % TRIM5 α + cell increase between the Minor and Major genotypes with the Major genotypes having a similar increase following TCR engagement.
5. Figure 2f. I find the RNA-seq analysis difficult to understand in the context of HIV-1 infection and other infectious diseases. Although there may be a relative increase in specific RNAs in the Minor versus Major genotypes, the shift suggested by the RNA-seq data is difficult to understand since most people have the Major allele. If true, how is it that those with the Major allele generally have functional autophagy? The authors should explain this finding. Also, HIV-1 associated neurocognitive impairment has been associated with mitochondrial dysfunction. Are those with the Minor genotype at greater risk of having HIV-1 associated CNS conditions?
6. Figure 3b. It is of interest that there is a decrease in Tcm cells in the Minor genotype group. Since Tcm are an important site of the resting HIV-1 reservoir, it would be of interest to measure the reservoir size in the Minor versus the Major genotype groups.
7. Figure 3h, 3i. Are these figures showing data from the Minor or Major genotypes? The figure legend says Major but figure seems to suggest Minor. In any event, both Major and Minor should be shown.
8. The differences observed with the different mTOR inhibitors is likely a dose effect. The authors should note why they used the doses of drugs studied.
9. In the Discussion, the authors should discuss the important role of autophagy during HIV-1 infection as several HIV-1 encoded proteins including Nef, Vpr and Tat have been identified as altering autophagy in order to promote viral infection.

Reviewer #2 (Remarks to the Author):

Summary: The authors describe a targeted, candidate gene SNP association and immune profiling study of HIV infected individuals genotyped for various markers in ATG16L1, a gene involved in autophagy. They perform a variety of ex vivo analysis aimed at assessing the role of autophagy in HIV infection and conclude that people carrying two copies of the minor allele at rs6861 have enhanced autophagy which acts to limit HIV progression. Overall the authors present compelling functional data (at least to my eye) that support the conclusion of the protective effect of autophagy on HIV progression. However, the genetic data that motivated these investigations is very limited.

Major:

A detailed critique of the functional assays is beyond the scope of my expertise, however, I have some concerns over the genetic findings. Firstly, the SNP selection criteria are poorly described. The authors state "Initially, nine SNPs within the ATG16L1 gene were identified that impacted the clinical course of HIV-1" but no data is given to support this claim. Next the authors filter this list down to 5 they tested in their cohort based on a genotype count threshold of 10 (of note the inversion of the greater than and less than symbol on line 107 was the cause of some confusion) which is not well rationalized. They then arrive at rs6861 as being the only variant that remains associated after multiple comparison correction but do not state their threshold or how they calculated it (the associations fall well below genome-wide significance). All of these details matter when assessing the validity of their claim of association.

As well, the description of the sample population is insufficient to assess the likelihood of common confounders in genetic studies. The ancestry of the participants and whether they were assessed for relatedness is particularly important here. This concern also applies to the healthy controls they use in their ex vivo work. Is it unclear how well matched the healthy blood donor population would be the MSM living with HIV.

Additionally, there are several large genome-wide association studies of viral load, CD4 decline and disease progress and to my knowledge none of them identified variants in ATG16L1. Are the authors arguing that the impact of rs6861 is specific to this population or that the variant is associated but does not pass the strict statistical standards of GWAS? If the latter, it would be good to know if rs6861 shows even nominal association in these other studies (which should be publicly available) with the same direction of effect the authors observe.

Finally, regarding the genetic analysis, the language used throughout the paper seems to suggest a causal relationship between rs6861 and enhanced autophagy rather than an association. However, there is no data presented nor any analysis done to suggest by what mechanism this may be acting. Indeed, the authors themselves state "Molecular characterization of the SNP was beyond the scope of this manuscript". The manuscript needs to be re-written to reflect that the function of rs6861 is unknown and could easily be tagging multiple functions effects not assessed in this study.

The lack of appropriate methodological detail is also apparent in the RNA-seq analysis section. For example, no software is listed for performing the read mapping and statements like "Principle component analysis revealed clear separation" leave the reader confused as to what type of analysis was done. Also, the parameters used for the GSEA analysis are not specified. In particular it is not mentioned what background gene set was used. From the

figure it appears they used all ~20,000 coding genes which could easily lead to false associations even that many would not be expressed in T cells. This is a well known bias in gene set analyses and needs to be considered (see PMID:15972284 17098774 28259142).

I do also have some concerns over the functional data, although I'll leave a detailed critique to the other reviewers. First, the interpretation of fig 2e seems mis-stated to me "TCR-engagement resulted in increased frequencies of TRIM5 α + cells in minor compared to major CD4+ T cells" (ln 146 - 7). The figure does show a (modestly) significant difference in %TRIM5 + cells after stimulation in rs6861(T) homozygotes but does not appear to show higher levels than in rs6861(C) homozygotes. Indeed two rs6861(C) homozygotes have the highest proportion of TRIM5+ cells. This should be re-stated and discussed in light of the author's hypothesis.

Second, it is not clearly specified when the PBMC samples from the HIV+ donors were collected. The methods state that the sample was collected at least 2 years after seroconversion and not more than 1 year before the onset of AIDS but this is quite a long window. It would be important to know that samples from donors with the different genotypes were sampled at approximately the same time during their infection course. This could further be supported by knowing the CD4 count and viral load for these groups at the time of PBMC collection.

Minor:

Throughout the manuscript the terminology used to refer to genotypes is clunky. Calling individuals with a genotype minor or major could lead to confusion. See for example phrases like "Minor CD4+ T helper cells". I suggest being precise e.g. using rs6861(T) homozygotes instead of minor and rs6861(C) homozygotes instead of major.

Supplementary table 2 is mis-labeled. It presents genotype counts, not frequencies. Also unclear why the genotypes are specified as A AB and B when the proper allele codes can be used.

Response to reviewer 1

This manuscript describes the effect of a specific polymorphism in *ATG16L1* (rs6861) on autophagy and its role in improving T-cell immunity in persons chronically infected with HIV-1. The investigators convincingly show that the rs6861 SNP functionally enhances autophagy and improves clinical outcomes including AIDS diagnosis and survival of those chronically infected with HIV-1. The work expands on the importance of autophagy in HIV-1 infection that had been largely unappreciated until the past decade.

Specific Comments

1. Figure 1 shows that there is a slight increase in the viral set-point and an increase in CD4 count for those with the Minor genotype. What are the intracellular levels of HIV-1 in the different genotypes?

- The Amsterdam Cohort Studies on HIV infection and AIDS is a prospective study among men who have sex with men (MSM) that started in 1984 (PMID: 21192230; PMID: 9382366). For this study, data and samples from HIV-1-infected MSM enrolled between October 1984 and March 1986 were used to determine the effect of the *ATG16L1* polymorphisms on the untreated course of HIV-1 infection. Intracellular viral load was not determined in these HIV-1 infected MSM. In this cohort of untreated HIV-1 infected individuals, viral load in the serum was determined using nucleic acid sequence-based amplification assays as a measure of viral replication (PMID: 8638160; PMID: 9412697; PMID: 21811574). Herewith, we show that *ATG16L1* rs6861(TT) genotyped individuals – i.e carrying the minor genotype – displayed a slightly decreased set-point plasma viral load ($p=0.0140$) and increased CD4 cell counts ($p=0.0361$) as compared to HIV-1 infected individuals carrying the rs6861(CC) or heterozygous rs6861(CT) genotypes (Revised manuscript: new Figure 2 c,d). We have clarified these findings and included additional relevant literature in the revised manuscript (lines 126-137 & 493-505).

2. Presumably, when some participants were diagnosed with HIV-1, they were started on ART and have survived. For those that are fully suppressed, what is the reservoir size of the Minor genotypes versus Heterozygous and Major genotypes? It would be of considerable interest if those with the Minor genotype have a smaller reservoir size.

- For this study, data and samples from untreated HIV-1-infected MSM enrolled between October 1984 and March 1986, and actively followed-up until May 1996, were used to determine the impact of *ATG16L1* rs6861 genetic variation on the natural course of untreated HIV-1 chronic disease. Plasma HIV-1 RNA (viral load) is used as a surrogate marker of viral replication (PMID: 8638160; PMID: 9412697; PMID: 21811574). None of the 304 genotyped MSM received effective ART upon diagnosis as this was not available at the time. Moreover, in the early days of ART, World Health Organization guidelines recommended to start treatment in adults when CD4 T cells counts in the blood were below 200 cells/uL, which is after AIDS diagnosis. We have now included this information and clarified it in the revised manuscript (lines 493-505).

3. In figure 2c, engagement of TCR using soluble CD3 and anti-CD28 results in increased autophagy in the Minor genotype. It would be of interest to determine if HIV-1 GU-rich ssRNA itself shows a similar increase in autophagy in the Minor group.

- As requested, we have now investigated the potential stimulatory effect of GU-rich single-stranded RNA derived from HIV-1 LTR on autophagy flux in genotyped T cells. To this end, PBMCs from either rs6861(TT) or rs6861(CC) genotyped healthy individuals were treated with GU-rich ssRNA40 (Invivogen) or negative control ssRNA41 (Invivogen) in the presence of autophagosome-lysosome fusion inhibitor bafilomycin, followed by intracellular staining for LC3-II in combination with immune lineage markers by flow cytometry analysis (*Rebuttal Figure 1*, please see below). Accumulation of intracellular LC3-II levels in ssRNA-treated cells in the presence of bafilomycin A1 is indicative of enhanced autophagy flux [PMID: 2679965; PMID: 27919079]. Our data show that ssRNA-40 did not significantly increase autophagy flux in neither CD4⁺ or CD8⁺ T cell compartments derived from either rs6861(TT) and rs6861(CC) genotyped individuals at any of the analyzed concentrations, as evident from the lack of LC3-II accumulation in ssRNA/bafilomycin A1-treated cells when compared to bafilomycin A1-treated cells (*Rebuttal Figure 1*, please see below). Notably, we observed an increase of autophagy flux in the CD3⁻ subpopulation (comprising mostly of monocytes, B cells and NK cells) upon treatment with ssRNA40, as compared to negative control ssRNA-41, in rs6861(TT) as well as rs6861(CC) genotyped PBMCs. The autophagy-activating effect of ssRNA-40 in the CD3⁻ population likely stems from the more prevalent TLR7 and TLR8 expression in B-cell and myeloid immune cell compartments, respectively [PMID: 11970999; PMID: 23761632]. We can include these novel data if requested.

Rebuttal Figure 1. HIV-1 GU-rich ssRNA does not increase autophagy flux in rs6861(TT) genotyped T cells. PBMCs from rs6861(TT) or rs6861(CC) genotyped healthy individuals were pre-treated for 20 hours with HIV-1 GU-rich ssRNA (0.8, 2, or 5 µg/mL ssRNA 40) or control ssRNA (5 µg/mL ssRNA-41) followed by 4h co-treatment with bafilomycin A1 (50 nM). PBMCs were then harvested and stained for LC3-II accumulation in combination with immune lineage markers (CD3, CD4, CD8) as described in the manuscript methods section (lines 564-575). **a** Flow cytometric representation of the gating strategy to determine LC3-II median fluorescence intensity (MFI) in rs6861(TT) genotyped PBMC-derived immune subsets as a measure of autophagy flux. **b-d** Autophagy flux upon treatment with ssRNA-40, as compared to ssRNA-41, determined by intracellular LC3-II fold increase (ssRNA+bafilomycin A1 treatment divided by bafilomycin A1 treatment only) in CD3⁺4⁺ T cells (**b**), CD3⁺8⁺ T cells (**c**) and CD3⁻ cells (**d**). Graphs depict medians ± interquartile range; n=4 donors per genotype. Statistical significance was determined using ANOVA with Dunnett's multiple comparison test to ssRNA-41 per genotype. * $p < 0.05$, *** $p < 0.001$

4. Figure 2e. Other than a single outlier in the Major genotype group, there appears to be very little difference in the % TRIM5 α + cell increase between the Minor and Major genotypes with the Major genotypes having a similar increase following TCR engagement.

- We agree with the reviewer. We have adjusted the manuscript to more accurately describe the TRIM5 α expression at steady-state and upon TCR engagement in genotyped individuals. At steady-state, we observed that rs6861(TT) genotyped CD4+ T cells displayed an increased TRIM5 α mean fluorescence intensity (MFI) as compared to rs6861(CC) (Revised manuscript, Figure 3d), but no difference in % TRIM5 α + CD4+ T cells between the genotypes was observed (Revised manuscript: Supplementary Figure 2a). Upon TCR stimulation, we observed that the % TRIM5 α + cells increased significantly in rs6861(TT) genotyped CD4+ T cells and less robustly in rs6861(CC) genotyped CD4+ T cells (Revised manuscript: Supplementary Figure 2a,b). As suggested, we have edited the text accordingly, and clarified the findings in the revised manuscript (lines 158-167, Figure 3d, Supplementary Figure 2a,b,c,d).

5. Figure 2f. I find the RNA-seq analysis difficult to understand in the context of HIV-1 infection and other infectious diseases. Although there may be a relative increase in specific RNAs in the Minor versus Major genotypes, the shift suggested by the RNA-seq data is difficult to understand since most people have the Major allele. If true, how is it that those with the Major allele generally have functional autophagy? The authors should explain this finding. Also, HIV-1 associated neurocognitive impairment has been associated with mitochondrial dysfunction. Are those with the Minor genotype at greater risk of having HIV-1 associated CNS conditions?

- We apologize for the confusion. Indeed, the majority (~90%, Revised manuscript: Figure 1a) of people are either homozygous major or heterozygotes for the rs6861 SNP. The heatmap depicted in Figure 2e (Revised manuscript) reflects the relative differences in expression levels of key autophagy genes (as compiled by Bordi et al, 2021 PMID: 34728604) between CD4+ T cells from genotyped individuals at steady-state (baseline). This RNA-seq analysis further supports our functional data represented in Figure 3a,b (Revised manuscript) that rs6861(TT) genotyped CD4+ T cells – i.e. homozygous for the minor allele – display enhanced levels of autophagy flux at steady-state when compared to rs6861(CC) genotyped individuals. This is exemplified by the combined relative increased expression of autophagy core genes (U2AF1, RAB9A, NEK9, MAP1LC3B2 and NBR1), lysosome-related genes (TMEM9, MAMBA, NAGA and LIPA), and genes associated with positive regulation of autophagy (ABL1, SIT5) in minor compared to major CD4+ T cells. Of note, our transcriptomic and functional analyses also indicate that also rs6861(CC) genotyped individuals – i.e. homozygous for the major allele – have functional autophagy. In Figure 3e (Revised manuscript) a smaller number of autophagy core genes (HDAC6, GOLGA2 and PMAIP1) are relatively increased in CD4+ T cells from major compared to minor genotyped individuals. Additionally, treatment with v-ATPase inhibitor bafilomycin resulted in an increase of intracellular LC3-II protein in either minor or major CD4+ T cells (Revised manuscript: Figure 3a), which indicates that both genotypes are equipped with functional autophagy flux processes and generation of autophagosomes. Nonetheless, rs6861(TT) genotyped CD4+ T cells displayed a higher level of autophagy activity as evident by a relative greater increment of LC3 MFI, when compared to rs6861(CC) individuals, at steady-state (Revised manuscript: Figure 3b) as well as upon TCR engagement (Revised manuscript: Figure 3c). These functional and transcriptomic autophagy analyses therefore underscore the intrinsically higher autophagy flux potential of minor genotyped individuals. We have included supplemental information regarding the RNA-seq analysis of autophagy-related transcriptome as well as clarified these findings in the manuscript (lines 168-182; 651-668).

Neurocognitive impairment has not been analyzed in detail in the Amsterdam Cohort Studies. However, we do know which participants have developed AIDS-related dementia as diagnosed by the clinician: 17 out of the 304 untreated HIV-1-infected individuals employed in the targeted SNP analysis

were diagnosed with AIDS-related dementia, of whom only 2 had the minor genotype. Thus, there is no apparent overrepresentation of the minor genotype - i.e. carrying the rs6861(TT) genetic variant - in this group of dementia-diagnosed untreated HIV-1 infected MSM.

6. Figure 3b. It is of interest that there is a decrease in Tcm cells in the Minor genotype group. Since Tcm are an important site of the resting HIV-1 reservoir, it would be of interest to measure the reservoir size in the Minor versus the Major genotype groups.

- As outlined in our response to reviewer's point 2, we are unable to determine the reservoir size in the untreated HIV-1 infected individuals from Amsterdam Cohort studies. While serum is considered to harbor recently produced virus variants with a short half-life of about 1.3h (PMID: 10449298), PBMCs are considered to harbor a combination of recently produced and archived virus variants (PMID: 9144289; PMID: 9671771). As the HIV-infected MSM in this study are treatment-naïve, viral load in the serum (plasma HIV-1 RNA) is used as a surrogate marker of active viral replication (PMID: 8638160; PMID: 9412697; PMID: 21811574). We have now included this information and clarified it in the revised manuscript (lines 493-505).

7. Figure 3h, 3i. Are these figures showing data from the Minor or Major genotypes? The figure legend says Major but figure seems to suggest Minor. In any event, both Major and Minor should be shown.

- We apologize for the confusion. To improve clarity, we have now also included in the Figure panel 4 h,i (Revised manuscript) that the data herewith depicts the beneficial impact of treatment with autophagy-enhancing therapeutics on the proliferative capacity and renewal of the lymph-node homing of major genotyped CD4+ T cell compartment – i.e carrying the rs6861(CC) genetic variant. As requested, we have now also included additional experimental data on the effect of autophagy-enhancing drug treatment on the immune functionality of minor genotyped CD4+ as well as CD8+ T cell compartments - i.e carrying the rs6861(TT) genetic variant – (Revised manuscript: Supplementary Figure 5a,b,c,d,e). Altogether, these data strongly suggest that pharmaceutically enhanced autophagy improved T cell responses in major genotyped individuals and to a much lesser extent in minor genotyped individuals. The latter further underscoring the higher levels of basal autophagy activity intrinsically present in the rs6861(TT) genotyped individuals as shown in Figure 3. As requested, we have now included these novel results and clarified it the revised manuscript (Figure 4h,l, Supplementary Figure 4d, Supplementary Figure 5a,b,c,d,e; lines 206-227; 267-270; 307-317)

8. The differences observed with the different mTOR inhibitors is likely a dose effect. The authors should note why they used the doses of drugs studied.

- The dosage of autophagy-enhancing drugs used herein were selected based on our previous work (PMID: 33637808) as well as on their capacity to induce autophagy flux and cell proliferation without affecting cell viability using the U87.LC3-mCherry-GFP reporter cells and flow cytometry-based Cell Trace and LIVE/DEAD viability assays, respectively. Non-toxic, autophagy-inducing, median concentrations within the tested range displaying proliferation-inducing capacities in genotyped CD4+ and CD8+ T cells were thereby selected: 100 uM carbamazepine, 5 nM everolimus, and 1 uM MK-2206. As requested, we have now included these additional experimental data in Supplementary Figure 4a,b,c,d and clarified it in the revised manuscript (lines 206-218).

9. In the Discussion, the authors should discuss the important role of autophagy during HIV-1 infection as several HIV-1 encoded proteins including Nef, Vpr and Tat have been identified as altering autophagy in order to promote viral infection.

- We thank the reviewer for the suggestion. As requested, we have now discussed the impact of HIV-1 accessory proteins *Nef*, *Vpr* and *Tat* on autophagy pathway upon HIV-1 infection (lines 392-395).

Response to reviewer 2

Summary: The authors describe a targeted, candidate gene SNP association and immune profiling study of HIV infected individuals genotyped for various markers in *ATG16L1*, a gene involved in autophagy. They perform a variety of ex vivo analyses aimed at assessing the role of autophagy in HIV infection and conclude that people carrying two copies of the minor allele at rs6861 have enhanced autophagy which acts to limit HIV progression. Overall the authors present compelling functional data (at least to my eye) that support the conclusion of the protective effect of autophagy on HIV progression. However, the genetic data that motivated these investigations is very limited.

Major:

A detailed critique of the functional assays is beyond the scope of my expertise, however, I have some concerns over the genetic findings.

Firstly, the SNP selection criteria are poorly described. The authors state “Initially, nine SNPs within the ATG16L1 gene were identified that impacted the clinical course of HIV-1” but no data is given to support this claim.

Next the authors filter this list down to 5 they tested in their cohort based on a genotype count threshold of 10 (of note the inversion of the greater than and less than symbol on line 107 was the cause of some confusion) which is not well rationalized.

They then arrive at rs6861 as being the only variant that remains associated after multiple comparison correction but do not state their threshold or how they calculated it (the associations fall well below genome-wide significance). All of these details matter when assessing the validity of their claim of association.

- We apologize for the confusion and for the incorrect placement of “<” symbol. We have now included our decision tree analyses with corresponding data tables in a new main Figure 1 in the revised manuscript.

Here we performed a targeted analysis of tagging SNPs in *ATG16L1* gene using available data from 304 untreated HIV-1 infected individuals. Nine *ATG16L1* tagging SNPs were investigated within the Amsterdam Cohort Studies using genotyping analyses performed with Illumina’s Infinium HumanHap 300 BeadChip (Revised manuscript: Figure 1a,b,c).

Next, four out of these nine *ATG16L1* tagging SNPs were excluded from subsequent analyses as less than 10 participants homozygous for the less common allele were available within the ACS cohort thus precluding meaningful comparison to the more common and heterozygous variants (Revised manuscript: Figure 1c, grey shaded).

Subsequently, for the remaining five *ATG16L1* SNPs, univariate Cox regression survival analysis revealed a significant protective effect for two *ATG16L1* SNPs (rs6861 and rs7563345) using AIDS diagnosis CDC 1987 as endpoint, and for four *ATG16L1* SNPs (rs2241880, rs6861, rs3792106 and rs7563345) using AIDS-related death as endpoint (Revised manuscript: Figure 1d). After Bonferroni

multiple testing correction (adjusted $p < 0.01$), only the protective effect of *ATG16L1* SNP rs6861 on both endpoints remained significant and was thereby identified to be the only investigated *ATG16L1* SNP to be associated with delayed HIV-1 disease progression (Revised manuscript: Figure 1d,e). We have now included all these results and clarified it in the revised manuscript (Figure 1 a,b,c,d,e; lines 104-124)

As well, the description of the sample population is insufficient to assess the likelihood of common confounders in genetic studies. The ancestry of the participants and whether they were assessed for relatedness is particularly important here. This concern also applies to the healthy controls they use in their ex vivo work. Is it unclear how well matched the healthy blood donor population would be to the MSM living with HIV.

- PBMCs from HIV-1 infected individuals were obtained via the Amsterdam Cohort Studies on HIV-1 in MSM (men who have sex with men), which consists primarily of white male participants. Eigenstrat, a method using principal component analysis to explicitly model ancestry differences among participants (implemented in Eigensoft) was used to identify outliers in the HIV-1 infected population who were subsequently removed from analysis. The first two eigenvectors were used as covariates to correct for any remaining population structure. Subsequently, genetic data from 304 HIV-1 infected MSM were used for SNP analyses (PMID:21811574). The identification of outliers and relevant literature has now been added in the Revised manuscript (lines 497-500).

Healthy PBMCs were obtained from buffy coats derived from blood donors supplied for research purposes by Sanquin Blood Supply Foundation (Amsterdam). These donations are anonymized and we do not have access to information (such as age, sex, relatedness) pertaining to the individual donors. General characteristics of the Dutch donor population have been published (PMID: 30590867) and this information and reference has been added to the revised manuscript (lines 528-529). Functional effect of SNP was evaluated in both healthy and HIV-1 infected individuals. Healthy donors included in this study are not matched to HIV-1 infected population and therefore no direct comparisons between these two groups were made in the manuscript.

As requested, we have now included a new Figure detailing the clinical characteristics of the selected PBMCs from HIV-1 infected MSM used for *ex vivo* work, which were matched between genotypes on sex, year of seroconversion, age at seroconversion, CD4 count and time between seroconversion, sample selection, and AIDS diagnosis (Revised manuscript: Supplementary Figure 3).

Additionally, there are several large genome-wide association studies of viral load, CD4 decline and disease progress and to my knowledge none of them identified variants in *ATG16L1*. Are the authors arguing that the impact of rs6861 is specific to this population or that the variant is associated but does not pass the strict statistical standards of GWAS? If the latter, it would be good to know if rs6861 shows even nominal association in these other studies (which should be publicly available) with the same direction of effect the authors observe.

- We thank the reviewer for this comment. Indeed, we should specify more in-depth why *ATG16L1* SNP rs6861 is not flagged in GWAS.

Here we performed targeted gene analysis to determine the effect of genetic variation in *ATG16L1* on HIV disease progression. The rs6861 SNP was not identified in the full GWA, which is not unanticipated as the non-progressive HIV-1 genetic determinants HLA-B*57 (HCP-5) and CCR5 $\Delta 32$ heterozygotes also do not pass the strict GWA statistical standards (PMID: 21811574, PMID: 19050382, PMID: 21051598). Furthermore, it should be noted that the targeted analysis of tagging SNPs in *ATG16L1* was not performed in extreme groups (i.e. progressors *versus* elite controllers).

Here, we used a survival analysis including the complete follow-up data from the entire seroconverter cohort irrespective of the disease course profile, which consists of HIV-1 controllers as well as HIV-1 progressors (ranging from fast-intermediate-slow progressors) as they occurred in our ACS study population. We have clarified this in the revised manuscript (lines 494-505).

Finally, regarding the genetic analysis, the language used throughout the paper seems to suggest a causal relationship between rs6861 and enhanced autophagy rather than an association. However, there is no data presented nor any analysis done to suggest by what mechanism this may be acting. Indeed, the authors themselves state “Molecular characterization of the SNP was beyond the scope of this manuscript”. The manuscript needs to be re-written to reflect that the function of rs6861 is unknown and could easily be tagging multiple functions effects not assessed in this study.

- We agree with the reviewer. We have adjusted the manuscript and study limitations sections to remove any apparent suggestion inferring direct causality between rs6861 SNP and enhanced autophagy activity described in this study and reflect that the rs6861 SNP function is yet unknown and follow-up studies are required to investigate the latter (Revised manuscript: lines 463-469; 481-491).

The lack of appropriate methodological detail as also apparent in the RNA-seq analysis section. For example, no software is listed for performing the read mapping and statements like “Principle component analysis revealed clear separation” leave the reader confused as to what type of analysis was done. Also, the parameters used for the GSEA analysis are not specified. In particular it is not mentioned what background gene set was used. From the figure it appears they used all ~20,000 coding genes which could easily lead to false associations even that many would not be expressed in T cells. This is a well known bias in gene set analyses and needs to be considered (see PMID:15972284 17098774 28259142).

- Our apologies for the lack of methodological data in regard to the RNA-seq. analyses performed. As requested, we have now revised the methods section of the manuscript to reflect in detail which analyses were done using which software and with which parameters. We hope this added information removes any confusion. Below an overview of tools, software and rationale used, all of which have been added to the revised manuscript (lines 624-675).

Library preparation, sequencing, read demultiplexing, mapping, and quantification of gene expression were all outsourced to QIAGEN. Library preparation was done using the QIAseq UPX 3' Transcriptomics Kit and yielded expression data from 45369 types of RNA. Quality control of the library preparation was performed using Agilent DNA 7500 Chips in combination with qPCR-based quantification. The library pool was sequenced on a NextSeq 500 sequencing instrument with 100 basepair (bp) read length for read 1 and 27 bp for read 2. Raw data was demultiplexed and FASTQ files were generated using the bcl2fastq Conversion Software from Illumina. Raw sequencing reads were demultiplexed according to sample indices using the “Demultiplex QIAseq UPX 3' reads” tool of QIAGEN's proprietary CLC Genomics Workbench (version 21.0.1). The “Quantify QIAseq UPX 3' workflow” (CLC Genomics Workbench, 21.0.1) was used to process the demultiplexed sequencing reads with default settings. The reads were then mapped to the human genome GRCh38 and annotated using the NCBI RefSeq GRCh38.p12 annotation. The “Empirical Analysis of DGE” algorithm (CLC Genomics Workbench, 21.0.1) was used for differential gene expression analysis with default settings. It is an implementation of the ‘Exact Test’ for two-group comparisons (PMID17728317) and incorporated in the EdgeR Bioconductor package (PMID19910308); a software package using a overdispersed Poisson model to account for biological and technical variability and empirical Bayes

methods to moderate the degree of overdispersion across transcripts. It is suitable even for minimal levels of replication. Principal component analysis was performed with ClustVis (PMID25969447) using counts per million of the 500 genes with the highest variance with unit variance scaling. GSEA was performed with clusterProfiler (PMID22455463). Genes that were actively transcribed in our samples (isolated CD4+ T cells) were selected as background gene set prior to GSEA. This was achieved by removing genes which did not have any expression for each comparison between the two genotypes, thus the background gene set was customized for each comparison based on genes that showed actual transcription (yielding ~21,000 to 22,000 genes per comparison). This way, the sample source bias as mentioned in PMID28259142 was reduced. GSEA was performed with the following parameters: only genes with expression used as background gene set (~21,000 to 22,000), minimum geneset size = 5 genes and maximum geneset size = 500 genes with a cutoff p -value = 0.05. Cells derived from rs6861(CC) genotype individuals were used as baseline. To determine basal differences in autophagy-related genes between genotypes, we employed the 'Autophagy Gene Toolbox' (34728604) consisting of 604 autophagy-related annotated genes. To evaluate differences in T-cell-related gene expression, a list of 262 genes related to T-cell immunology and inflammation was curated by mining the GO Biological Process gene set library (c5.go.bp.v7.4) using the terms "T cell", "T helper", "Interleukin", "Interferon gamma", "Chemokine", "Inflammasome", and "Antigen".

I do also have some concerns over the functional data, although I'll leave a detailed critique to the other reviewers. First, the interpretation of fig 2e seems mis-stated to me "TCR-engagement resulted in increased frequencies of TRIM5 α + cells in minor compared to major CD4+ T cells" (ln 146 - 7). The figure does show a (modestly) significant difference in %TRIM5 + cells after stimulation in rs6861(T) homozygotes but does not appear to show higher levels than in rs6861(C) homozygotes. Indeed two rs6861(C) homozygotes have the highest proportion of TRIM5+ cells. This should be re-stated and discussed in light of the author's hypothesis.

- We agree with the reviewer. Indeed, our choice of phrasing did not appropriately reflect the data. We have now adjusted the manuscript to more accurately describe the TRIM5 α expression upon TCR engagement in genotyped individuals. In brief, upon TCR stimulation, we observed that the frequency (%) of TRIM5 α + cells increased significantly in rs6861(TT) genotyped CD4+ T cells compared to steady-state, while this increase was less robust in rs6861(CC) genotyped CD4+ T cells (Revised manuscript: Supplementary Figure 2a,b). As suggested, we have edited the text accordingly, and clarified the findings in the revised manuscript (lines 158-167, Figure 3d, Supplementary Figure 2a,b,c,d).

Second, it is not clearly specified when the PBMC samples from the HIV+ donors were collected. The methods state that the sample was collected at least 2 years after seroconversion and not more than 1 year before the onset of AIDS but this is quite a long window. It would be important to know that samples from donors with the different genotypes were sampled at approximately the same time during their infection course. This could further be supported by knowing the CD4 count and viral load for these groups at the time of PBMC collection.

- As requested, we have now detailed the clinical characteristics of the selected PBMCs from HIV-1 infected donors during the chronic phase of infection. PBMCs with the different genotypes were matched on sex, year of seroconversion, age at seroconversion, CD4 count and time between seroconversion, sample selection, and AIDS diagnosis (Revised manuscript: Supplementary Figure 3).

Minor:

Throughout the manuscript the terminology used to refer to genotypes is clunky. Calling individuals with a genotype minor or major could lead to confusion. See for example phrases like "Minor CD4+ T helper cells". I suggest being precise e.g. using rs6861(T) homozygotes instead of minor and rs6861(C) homozygotes instead of major.

- We thank the reviewer for the suggestion. As requested, we have adjusted the terminology and used throughout the revised manuscript 'rs6861(TT)' when referring to homozygous for the minor allele and 'rs6861(CC)' when referring to homozygous for the major allele. We have outlined the genotypes terminology in the revised manuscript (Figure 1e, lines 90-99).

Supplementary table 2 is mis-labeled. It presents genotype counts, not frequencies. Also unclear why the genotypes are specified as A AB and B when the proper allele codes can be used.

- We agree with the reviewer. As requested, we have now incorporated this table into Figure 1c (Revised manuscript) and have edited its description accordingly. We have removed the 'A, AB, and B labels' and instead have opted for 'homozygous' and 'heterozygous' labeling combined with a specification of the change in nucleotide per SNP per variant.

REVIEWER COMMENTS

Reviewer #2 (Remarks to the Author):

I appreciate the authors' detailed response to my initial review and the thorough additions they have made to the manuscript. Although I am largely satisfied with their efforts, I am not fully satisfied with the response to my third comment regarding the reproducibility of the association at rs6861. To my knowledge there are several published genome-wide studies of HIV progression with similar population demographics and progression phenotypes that could be accessed to assess whether this association can be replicated independently (PMID: 2004116, 20064070, 21502085). Indeed, in the authors' response they cite van Manen et al (PMID 21811574), an ACS-lead paper which used data from the French Genomics of Resistance to Immunodeficiency Virus cohort as a replication set. This suggests to me that assessment of replication can be accomplished without undue additional effort. If this effect of rs6861 can be independently replicated, that would greatly enhance my confidence in the association.

Response to reviewer 2

I appreciate the authors' detailed response to my initial review and the thorough additions they have made to the manuscript. Although I am largely satisfied with their efforts, I am not fully satisfied with the response to my third comment regarding the reproducibility of the association at rs6861. To my knowledge there are several published genome-wide studies of HIV progression with similar population demographics and progression phenotypes that could be accessed to assess whether this association can be replicated independently (PMID: 2004116, 20064070, 21502085). Indeed, in the authors' response they cite van Manen et al (PMID 21811574), an ACS-lead paper which used data from the French Genomics of Resistance to Immunodeficiency Virus cohort as a replication set. This suggests to me that assessment of replication can be accomplished without undue additional effort. If this effect of rs6861 can be independently replicated, that would greatly enhance my confidence in the association.

In this study, participants from the Amsterdam Cohort Studies on HIV infection and AIDS (ACS) were included without any pre-selection for the disease course profile (fast progressors *versus* slow/non progressors). We included all (HIV-infected) MSM participants that were enrolled in the cohort in 1984-1985 irrespective of follow-up time or clinical characteristics (asymptomatic *versus* symptomatic). There are other genome-wide cohort studies of HIV progression such as GRIV, MACS and Euro-CHAVI cohorts. However, it is important to note that all these available cohorts, contrastingly to ACS cohort, perform analyses in extreme patient groups with pre-determined clinical characteristics, and are thereby not comparable. Furthermore, in our manuscript we build up on the cohort genetic and clinical data with in-depth functional analyses in PBMCs derived from chronically HIV-1 infected individuals. The immune phenotyping and immune functionality analyses *ex vivo* thereby strengthens the association of the *ATG16L1* genotype with phenotypically improved clinical disease outcomes and superior CD4+ and CD8+ T cell immunity in chronic HIV-1 infection.

To further clarify the divergence between the cohorts, we have provided below detailed information outlining the population demographics and progression phenotypes of the available HIV progression cohorts, which underscore our statement that these cohorts are not fitted as replication sets:

- i) Genomics of Resistance to Immunodeficiency Virus cohort (GRIV, PMID: 9091061, PMID: 19754311) comprises of male and female HIV-1 infected individuals who contracted infection via different modes of transmission (homosexual or heterosexual contact, intravenous drug usage or blood transfusion). In the GRIV cohort, patients were selected based on their disease course profile with enrichment for either slow/non-progressors or fast progressors, with the aim to use fast progressors as a mirror control population to non-progressors. In van Manen *et al* (PMID 21811574) study, the authors investigated the top-ranked non-genome-wide significant SNP associations ($p < 0.00001$). Herein, it was reported that specifically 5 SNPs out of the 20 top-ranked analyzed SNPs within ACS cohort were associated with disease progression profiles in the GRIV cohort. The *ATG16L1* rs6861 SNP is not present in the GRIV cohort.
- ii) Multicenter AIDS Cohort Study (MACS, PMID: 20064070), similarly to GRIV cohort, performed analysis on HIV-1 disease progression rates using extreme ends of this phenotypic distribution (rapid progressors *versus* long-term non-progressors). In the MACS cohort, study participants were thereby selected based on their progression phenotypes with enrichment on three pre-set

categories of AIDS-free interval (rapid progressors *versus* moderate progressors *versus* long-term non-progressors). Hence, MACS and ACS cohort diverge in their patient inclusion and follow up analyses and non-significant differences of the *ATG16L1* genotype in the MACS extreme patient groups were observed.

- iii) Center for HIV-AIDS Vaccine Immunology (Euro-CHAVI, PMID: 17641165) performed whole-genome associations using set-point viral load (plasma HIV RNA) as endpoint during the asymptomatic phase of the infection. In the Euro-CHAVI cohort, patients were thereby selected based on their disease clinical profile with only individuals displaying long-term spontaneous control of viral load admitted to the study. Within the ACS cohort, *ATG16L1* rs6861(TT) genotyped HIV-1 infected individuals displayed a decreased set-point plasma viral load ($p = 0.0140$, Figure 2c). Nevertheless, we observed considerable overlap of viral load data points between rs6861(TT) and rs6861(CC)/(CT) genotypes in the ACS cohort and non-significant differences in the Euro-CHAVI asymptomatic genotyped patients, which further support our observation that *ATG16L1* rs6861(TT) genotyped individuals do not spontaneously control viral load as reported for example for the non-progressive genetic determinants HLA-B57 (HCP-5) and CCR5 $\Delta 32$ heterozygotes (PMID: 21811574, PMID: 19050382, PMID: 21051598). With the ACS cohort, we were able to show that the prolonged control of HIV-1 disease pathogenesis in rs6861(TT) genotyped individuals using additional clinical endpoints (AIDS diagnosis and AIDS-related death) correlated *ex vivo* with superior antiviral T cell immune functionality in chronically HIV-1 infected individuals.

ACS cohort thus particularly focuses on the natural history of HIV-1 disease, including the complete follow-up data from the entire seroconverter cohort irrespective of clinical symptoms, viral load control and progression phenotypes (ranging from fast-intermediate-slow progressors) as they naturally occurred in our ACS study population, which also permitted us to analyze the impact of genetic variation in several host dependency/antiviral factors on the clinical course of untreated HIV-1 infection (PMID: 28024153, PMID: 32287057, PMID:25162766). Our manuscript derives from our discovery that *ATG16L1*-mediated autophagy represents an antiviral mechanism on acute mucosal HIV-1 infection (PMID: 27919079). Herein, we demonstrated that the *ATG16L1* rs6861(TT) genotype correlated clinically with improved AIDS-free survival of untreated HIV-1-infected individuals without spontaneous control of viral load, and functionally with augmented CD4+ as well as CD8+ T-cell immunity as evidenced by increased proliferation, sustained immune responsiveness, and suppressed exhaustion/immunosenescence signatures in chronically HIV-1 infected individuals *ex vivo* (Figure 2a-d, Figure 4f,g; Figure 5a-d; Figure 6d-g). Importantly, the in-depth immune phenotyping and functional analyses in chronically HIV-1 infected individuals thereby strengthens the association of *ATG16L1* rs6861 SNP with improved clinical disease outcomes and T-cell immune mediated HIV-1 control in chronic HIV-1 infection.

REVIEWERS' COMMENTS

Reviewer #2 (Remarks to the Author):

I appreciate the authors' detailed response. However, the lack of replication of the genetic associations reported in this manuscript in other studies remains a concern. While the authors argue that the phenotype captured in the ACS is fundamentally different from other studies, I believe the genetic underpinnings of HIV progression should be consistent across the spectrum. The extreme phenotype design, as suggested in PMID: 26350511, should provide more power to detect signals compared to the full distribution.

In the second Euro-CHAVI paper (20041166), Fellay et al. tested progression phenotypes across the spectrum of low and high viral load in 2,554 PLWH and, to my knowledge, did not observe associations at ATG16L1. Additionally, the authors demonstrated an association between rs6861 and setpoint viral load in the ACS, supporting the hypothesis that these phenotypes are similar. Considering this, I would have expected to see replication in larger studies of setpoint viral load.

Response to reviewer 2

I appreciate the authors' detailed response. However, the lack of replication of the genetic associations reported in this manuscript in other studies remains a concern. While the authors argue that the phenotype captured in the ACS is fundamentally different from other studies, I believe the genetic underpinnings of HIV progression should be consistent across the spectrum. The extreme phenotype design, as suggested in PMID: 26350511, should provide more power to detect signals compared to the full distribution.

- In this study, we show that the prolonged control of HIV-1 disease pathogenesis in *ATG16L1* rs6861(TT) genotyped individuals using clinical endpoints (AIDS diagnosis and AIDS-related death) and *ex vivo* functional analyses (T cell immune phenotyping and immune functionality in chronically HIV-1 infected individuals) is not consistent with spontaneous control of viral load as reported for other non-progressive genetic determinants of HIV progression profiles (PMID: 19050382, PMID: 21051598). We have now included in the study limitations section of the revised manuscript the unfeasibility to replicate the *ATG16L1* phenotype identified with ACS cohort due to the divergence between available cohort studies design and analyses (page 12, revised manuscript). We would like to thank the reviewer for sharing the study design suggestion for rare variant association studies (PMID: 26350511), such as those testing alleles with a minor allele frequency (MAF) of less than 0.05. The herewith described *ATG16L1* rs6861 SNP has a MAF of 0.09 (Figure 1d, Revised manuscript).

*In the second Euro-CHAVI paper (20041166), Fellay et al. tested progression phenotypes across the spectrum of low and high viral load in 2,554 PLWH and, to my knowledge, did not observe associations at *ATG16L1*. Additionally, the authors demonstrated an association between rs6861 and setpoint viral load in the ACS, supporting the hypothesis that these phenotypes are similar. Considering this, I would have expected to see replication in larger studies of setpoint viral load.*

- The *ATG16L1* rs6861(TT) genotyped HIV-1 infected individuals displayed a decreased setpoint viral load ($p = 0.0140$, Figure 2c) with considerable overlap of viral load data points between rs6861(TT) and rs6861(CC)/(CT) genotypes, which further support our observation that the increased containment of chronic HIV-1 disease in *ATG16L1* rs6861(TT) genotyped individuals is likely not primarily driven by impact on viraemia but rather stemming from a different protective mechanism modulating HIV disease progression (as proposed for the gene variants associated with HIV control and located around the *ZNDR1* and *RNF39* genes in PMID: 20041166). In fact, our *ex vivo* functional analyses substantiates the associations of the clinical disease phenotype of *ATG16L1* rs6861(TT) gene variant with longer-lasting CD4+ and CD8+ T-cell immune mediated HIV-1 control in chronically HIV-1 infected individuals.